# WAVELET-BASED GRAPH CONVOLUTION VIA CHEBYSHEV DECOMPOSITION

## ABSTRACT

Spectral graph convolution, an important tool of data filtering on graphs, relies on two essential decisions: selecting spectral bases for signal transformation and parameterizing the kernel for frequency analysis. While recent techniques mainly focus on standard Fourier transform and vector-valued spectral functions, they fall short in flexibility to model signal distributions over large spatial ranges, and capacity of spectral function. In this paper, we present a novel wavelet-based graph convolution network, namely WaveGC, which integrates multi-resolution spectral bases and a matrix-valued filter kernel. Theoretically, we establish that WaveGC can effectively capture and decouple short-range and long-range information, providing superior filtering flexibility, surpassing existing graph convolutional networks and graph Transformers (GTs). To instantiate WaveGC, we introduce a novel technique for learning general graph wavelets by separately combining odd and even terms of Chebyshev polynomials. This approach strictly satisfies wavelet admissibility criteria. Our numerical experiments showcase the capabilities of the new network. By replacing the Transformer part in existing architectures with WaveGC, we consistently observe improvements in both short-range and long-range tasks. This underscores the effectiveness of the proposed model in handling different scenarios. Our code is available at `https://anonymous.4open.science/r/WaveGC`.

## 1 INTRODUCTION

Spectral graph theory (SGT) (Chung, 1997), which enables analysis and learning on graph data, has firmly established itself as a pivotal methodology in graph machine learning. A significant milestone in SGT is the generalization of the convolution operation to graphs, as convolution for grid-structured data, i.e. sequences and images, has demonstrated remarkable success (LeCun et al., 1998; Hinton et al., 2012; Krizhevsky et al., 2012). Significant research interests in graph convolution revolve around two key factors: (1) *designing diverse bases for spectral transform*, and (2) *parameterizing powerful graph kernel*. For (1), the commonly used graph Fourier basis, consisting of the eigenvectors of the graph Laplacian (Shuman et al., 2013), stands as a prevalent choice. However, graph wavelets (Hammond et al., 2011) offer enhanced flexibility by constructing adaptable bases. For (2), classic approaches involve diagonalizing the kernel with fully free parameters (Bruna et al., 2013) or employing various polynomial approximations such as Chebyshev (Defferrard et al., 2016) and Cayley (Levie et al., 2018) polynomials. Additionally, convolution with a matrix-valued kernel serves as the spectral function of Transformer (Vaswani et al., 2017) under the shift-invariant condition (Li et al., 2021; Guibas et al., 2021).

Despite the existence of techniques in each aspect, the integration of these two lines into a unified framework remains challenging, impeding the full potential of graph convolution. In an effort to unravel this challenge, we introduce a novel operation — **Wave**let-based **G**raph **C**onvolution (WaveGC), which seamlessly incorporates both spectral basis and kernel considerations. In terms of spectral basis design, WaveGC is built upon graph wavelets, allowing it to capture information across the entire graph through a multi-resolution approach from highly adaptive construction of multiple graph wavelet bases. For filter parameterization, we opt for a matrix-valued spectral kernel with weight-sharing. Beyond adjusting diagonal frequencies, the matrix-valued kernel offers greater flexibility to spectral filtering, thanks to its larger parameter space.

To comprehensively explore WaveGC, we theoretically analyse and assess its information-capturing capabilities. In contrast to the K-hop basic message-passing framework, WaveGC is demonstrated to exhibit both significantly larger and smaller receptive fields concurrently, achieved through the manipulation of scales. Previous graph wavelet theory (Hammond et al., 2011) only verifies the localization in small scale limit. Instead, our proof is complete as it covers both extremely small and large scales from the perspective of information mixing (Di Giovanni et al., 2023). Moreover, our proof also implies that WaveGC is capable of simultaneously capturing both short-range and long-range information for each node, akin to (graph) Transformers, which facilitate global node interaction. Remarkably, WaveGC can distinguish information across diverse distances, thereby extending its flexibility beyond the scope of traditional Transformers.

To implement WaveGC, a critical step lies in constructing graph wavelet bases that satisfy two fundamental criteria: (1) meeting the wavelet admissibility criteria (Mallat, 1999) and (2) showing adaptability to different graphs. Existing designs of graph wavelets face limitations, with some falling short in ensuring the criteria (Xu et al., 2019a; 2022), while others having fixed wavelet forms, lacking adaptability (Zheng et al., 2021; Cho et al., 2023). To address these limitations, we propose an innovative and general implementation of graph wavelets. Our solution involves *approximating scaling function basis and multiple wavelet bases using odd and even terms of Chebyshev polynomials, respectively*. This approach is inspired by our observation that, after a certain transformation, even terms of Chebyshev polynomials strictly satisfy the admissibilitywe criteria, while odd terms supplement direct current signals. Through the combination of these terms via learnable coefficients, we aim to theoretically approximate scaling function and multiple wavelets with arbitrary complexity and flexibility. Our contributions are:

- We derive a new Wavelet-based graph convolution (WaveGC), which integrates multi-resolution bases and matrix-valued kernel, enhancing spectral convolution on large spatial ranges.
- We theoretically prove that WaveGC can capture and distinguish the information from short and long ranges, surpassing conventional graph convolutions and GTs.
- We pioneer a general implementation of learnable graph wavelets, employing odd terms and even terms of Chebyshev polynomials individually. This implementation strictly satisfies the wavelet admissibility criteria.
- Integrating WaveGC into three successful GTs as base models, our approach consistently outperforms base methods on both short-range and long-range tasks, achieving up to 26.20% improvement on CoraFull and 9.21% on VOC datasets.

## 2 PRELIMINARIES

An undirected graph can be presented as $\mathcal{G} = (\mathcal{V}, E)$, where $\mathcal{V}$ is the set of $N$ nodes and $E \subseteq \mathcal{V} \times \mathcal{V}$ is the set of edges. The adjacency matrix of this graph is $\boldsymbol{A} \in \{0, 1\}^{N \times N}$, where $\boldsymbol{A}_{ij} \in \{0, 1\}$ denotes the relation between nodes $i$ and $j$ in $\mathcal{V}$. The degree matrix is $\boldsymbol{D} = \text{diag}(d_1, \ldots . d_N) \in \mathbb{R}^{N \times N}$, where $d_i = \sum_{j \in \mathcal{V}} \boldsymbol{A}_{ij}$ is the degree of node $i \in \mathcal{V}$. The node feature matrix is $\boldsymbol{X} = [x_1, x_2, \ldots, x_N] \in \mathbb{R}^{N \times d_0}$, where $x_i$ is a $d_0$ dimensional feature vector of node $i \in \mathcal{V}$. Let $\hat{\boldsymbol{A}} = \boldsymbol{D}^{-\frac{1}{2}} \boldsymbol{A} \boldsymbol{D}^{-\frac{1}{2}}$ be the symmetric normalized adjacency matrix, then $\hat{\mathcal{L}} = \boldsymbol{I_n} - \hat{\boldsymbol{A}} = \boldsymbol{D}^{-\frac{1}{2}} (\boldsymbol{D} - \boldsymbol{A}) \boldsymbol{D}^{-\frac{1}{2}}$ is the symmetric normalized graph Laplacian. With eigen-decomposition, $\hat{\mathcal{L}} = \boldsymbol{U} \boldsymbol{\Lambda} \boldsymbol{U}^\top$, where $\boldsymbol{\Lambda} = \text{diag}(\lambda_1, \ldots, \lambda_N) \in \mathbb{R}^{N \times N}$ and $\boldsymbol{U} = [\boldsymbol{u_1}^\top, \ldots, \boldsymbol{u_N}^\top] \in \mathbb{R}^{N \times N}$ are the eigenvalues and eigenvectors of $\hat{\mathcal{L}}$, respectively. Given a signal $f \in \mathbb{R}^N$ on $\mathcal{G}$, the graph Fourier transform (Shuman et al., 2013) is defined as $\hat{f} = \boldsymbol{U}^\top f \in \mathbb{R}^N$, and its inverse is $f = \boldsymbol{U} \hat{f} \in \mathbb{R}^N$.

**Spectral graph wavelet transform (SGWT).** Hammond et al. (2011) redefine the wavelet basis (Mallat, 1999) on vertices in the spectral graph domain. Specifically, the SGWT is composed of three components: (1) *Unit wavelet basis*, denoted as $\Psi$ such that $\Psi = g(\hat{\mathcal{L}}) = \boldsymbol{U} g(\boldsymbol{\Lambda}) \boldsymbol{U}^\top$, where $g$ acts as a band-pass filter $g : \mathbb{R}^+ \to \mathbb{R}^+$ meeting the following *wavelet admissibility criteria* (Mallat, 1999):

$$\mathcal{C}_\Psi = \int_{-\infty}^{\infty} \frac{|g(\lambda)|^2}{|\lambda|} d\lambda < \infty. \tag{1}$$

To meet this requirement, $g(\lambda = 0) = 0$ and $\lim_{\lambda \to \infty} g(\lambda) = 0$ are two essential prerequisites. (2) *Spatial scales*, a series of positive real values $\{s_j\}$ where distinct values of $s_j$ with $\Psi_{s_j} =$

$Ug(s_j\Lambda)U^\top$ can control different size of neighbors. (3) *Scaling function basis*, denoted as $\Phi$ such that $\Phi = Uh(\lambda)U^\top$. Here, the function of $h : \mathbb{R}^+ \to \mathbb{R}^+$ is to supplement direct current (DC) signals at $\lambda = 0$, which is omitted by all wavelets $g(s_j\lambda)$ since $g(0) = 0$. Next, given a signal $f \in \mathbb{R}^N$, the formal SGWT (Hammond et al., 2011) is:

$$W_f(s_j) = \Psi_{s_j} f = Ug(s_j\Lambda)U^\top f \in \mathbb{R}^N, \tag{2}$$

where $W_f(s_j)$ is the wavelet coefficients of $f$ under scale $s_j$. Similarly, scaling function coefficients are given by $S_f = \Phi f = Uh(\Lambda)U^\top f \in \mathbb{R}^N$. Let $G(\lambda) = h(\lambda)^2 + \sum_j g(s_j\lambda)^2$, then if $G(\lambda) \equiv 1$, $\forall \lambda \in \Lambda$, the constructed graph wavelets are known as *tight frames*, which guarantee energy conservation of the given signal between the original and the transformed domains (Shuman et al., 2015). More spectral graph wavelets are introduced in Appendix E.

## 3    FROM GRAPH CONVOLUTION TO GRAPH WAVELETS

Spectral graph convolution is a fundamental operation in the field of graph signal processing (Shuman et al., 2013). Specifically, given a signal matrix (or node features) $X \in \mathbb{R}^{N \times d}$ on graph $\mathcal{G}$, the spectral filtering of this signal is defined with a kernel $\kappa \in \mathbb{R}^{N \times N}$ by the convolution theorem (Arfken, 1985):

$$\kappa *_{\mathcal{G}} X = \mathcal{F}^{-1}(\mathcal{F}(\kappa) \cdot \mathcal{F}(X)) \in \mathbb{R}^{N \times N}, \tag{3}$$

where $\cdot$ is the matrix multiplication operator, $\mathcal{F}(\cdot)$ and $\mathcal{F}^{-1}(\cdot)$ are the spectral transform (e.g., graph Fourier transform (Bruna et al., 2013)) and corresponding inverse transform, respectively. To implement a spectral convolution, two critical choices must be considered in Eq. (3): 1) the selection of the transform $\mathcal{F}$ and 2) the parameterization of the kernel $\kappa$.

### 3.1    GENERAL SPECTRAL WAVELET VIA CHEBYSHEV DECOMPOSITION

For the selection of the spectral transform $\mathcal{F}$ and its inverse $\mathcal{F}^{-1}$, it can be tailored to the specific nature of data. For set data, the Dirac Delta function (Oppenheim et al., 1997) is employed, while the fast Fourier Transform (FFT) proves efficient for both sequences (Li et al., 2021) and grids (Guibas et al., 2021). In the context of graphs, the Fourier transform ($\mathcal{F} \to U^\top$) emerges as one classical candidate. However, some inherent flaws limit the capacity of Fourier bases. (1) Standard graph Fourier bases, represented by one fixed matrix $U^\top$, maintain a constant resolution and fixed frequency modes. (2) Fourier transform lacks the adaptability to be further optimized according to different datasets and tasks. Therefore, *multiple resolution* and *adaptability* are two prerequisites for the design of an advanced base.

Notably, wavelet base is able to conform the above two demands, and hence offers enhanced filtering compared to Fourier base. For the resolution, the use of different scales $s_j$ allows wavelet to analyze detailed components of a signal at different granularity. More importantly, due to its strong spatial localization (Hammond et al., 2011), each wavelet corresponds to a signal diffused away from a central node (Xu et al., 2019a). Therefore, these scales also control varying receptive fields in spatial space, which enables the simultaneous fusion of short- and long-range information. For the adaptability, graph wavelets offer the flexibility to adjust the shapes of wavelets and scaling function. These components can be collaboratively optimized for the alignment of basis characteristics with different datasets, potentially enhancing generalization performance.

Next, we need to determine the form of the scaling function basis $\Phi = Uh(\Lambda)U^\top$, the unit wavelet basis $\Psi = Ug(\Lambda)U^\top$, and the scales $s_j$. The forms of $h$ and $g$ are expected to be powerful enough and easily available. Concurrently, $g$ should strictly satisfy the wavelet admissibility criteria, i.e., Eq. (1), and $h$ should complementarily provide DC signals. To achieve this target, we *separately introduce odd terms and even terms from Chebyshev polynomials (Hammond et al., 2011)* into the approximation of $h$ and $g$. Please recall that the Chebyshev polynomial $T_k(y)$ of order $k$ may be computed by the stable recurrence relation $T_k(y) = 2yT_{k-1}(y) - T_{k-2}(y)$ with $T_0 = 1$ and $T_1 = y$. After the following transform, we surprisingly observe that these transformed terms match all above expectations:

$$T_k(y) \to 1/2 \cdot (-T_k(y - 1) + 1). \tag{4}$$

To give a more intuitive illustration, we present the spectra of first six Chebyshev polynomials before and after the transform in Fig. 1 (a), where the set of odd and even terms after the transform

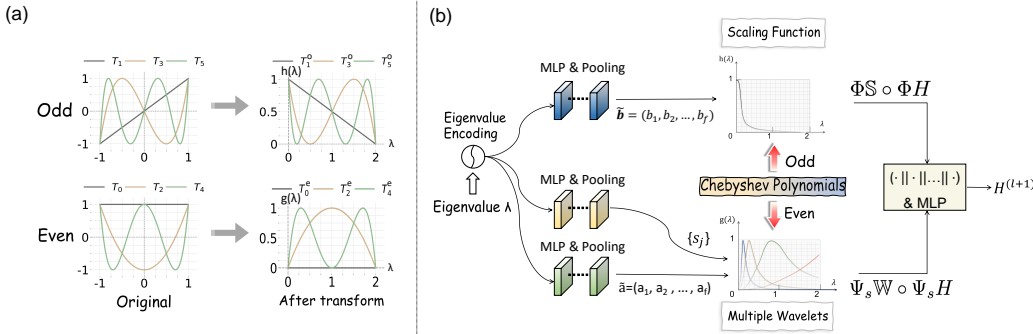

Figure 1: (a)An illustration of Chebyshev polynomials before and after the given transform. In this representation, we distinguish odd and even terms, presenting only the first three terms for each. (b)An overview of our proposed WaveGC.

are denoted as $\{T_i^o\}$ and $\{T_i^e\}$, respectively. From the figure, $g(\lambda = 0) \equiv 0$ for all $\{T_i^e\}$, and $h(\lambda = 0) \equiv 1$ for all $\{T_i^o\}$. Consequentially, $\{T_i^e\}$ and $\{T_i^o\}$ strictly meet the criteria and naturally serve as the basis of unit wavelet and scaling function. Moreover, not only can we easily get each Chebyshev term via iteration, but the constructed wavelet owns arbitrarily complex waveform because of the combination of as many terms as needed. Given $\{T_i^e\}$ and $\{T_i^o\}$, all we need to do is just to learn the coefficients to form the corresponding $g(\lambda)$ and $h(\lambda)$:

$$g(\mathbf{\Lambda}) = \sum_i^\rho a_i T_i^e(\mathbf{\Lambda}) \in \mathbb{R}^{N \times N}, \quad h(\mathbf{\Lambda}) = \sum_i^\rho b_i T_i^o(\mathbf{\Lambda}) \in \mathbb{R}^{N \times N}, \qquad (5)$$

where $\rho = K/2$ ($K$ is the total number of truncated Chebyshev terms), $\tilde{\boldsymbol{a}} = (a_1, a_2, \ldots, a_\rho) \in \mathbb{R}^{1 \times \rho}$ and $\tilde{\boldsymbol{b}} = (b_1, b_2, \ldots, b_\rho) \in \mathbb{R}^{1 \times \rho}$ represent two learnable coefficient vectors as follows:

$$\tilde{\boldsymbol{a}} = \text{Mean}(\boldsymbol{W_a}\hat{\boldsymbol{Z}} + \boldsymbol{b_a}), \quad \tilde{\boldsymbol{b}} = \text{Mean}(\boldsymbol{W_b}\hat{\boldsymbol{Z}} + \boldsymbol{b_b}), \qquad (6)$$

where $\{\boldsymbol{W_a}, \boldsymbol{W_b}\} \in \mathbb{R}^{d \times \rho}$ and $\{\boldsymbol{b_a}, \boldsymbol{b_b}\} \in \mathbb{R}^{1 \times \rho}$ are learnable parameters, and $\hat{\boldsymbol{Z}}$ is the eigenvalue embedding composed by the module in (Bo et al., 2023). Further details can be found in Appendix C. Also, we can learn the scales $\tilde{\boldsymbol{s}} = (s_1, s_2, \ldots, s_J)$ in the same way:

$$\tilde{\boldsymbol{s}} = \sigma(\text{Mean}(\boldsymbol{W_s}\hat{\boldsymbol{Z}} + \boldsymbol{b_s})) \cdot \overline{\boldsymbol{s}} \in \mathbb{R}^{1 \times J}, \qquad (7)$$

where $\sigma$ is sigmoid function, $\boldsymbol{W_s} \in \mathbb{R}^{d \times J}$ and $\boldsymbol{b_s} \in \mathbb{R}^{1 \times J}$ are learnable parameters, and $\overline{\boldsymbol{s}} = (\overline{s_1}, \overline{s_2}, \ldots, \overline{s_J})$ is a pre-defined vector to control the size of $\tilde{\boldsymbol{s}}$.

Based on our construction, $g(\lambda)$ is a strict band-pass filter in [0, 2], while $s$ can scale its shape in $g(s\lambda)$. Specifically, $s < 1$ "stretches" the shape of $g(\lambda)$, and $s > 1$ "squeezes" its shape. To maintain the same spectral interval [0, 2], we truncate $g(s\lambda)$ within $0 \leq s\lambda \leq 2$, or $0 \leq \lambda \leq 2/s$.

## 3.2 MATRIX-VALUED KERNEL VIA WEIGHT SHARING

Convolutional kernel $\mathcal{F}(\boldsymbol{\kappa})$ in Eq. (3) is usually parameterized in two ways: (1) *Vector-valued operator* such as $\text{diag}(\theta_\lambda)$ where diagonal elements form a parametrized function of the spectrum of the graph $\mathcal{G}$ (Bruna et al., 2013; Defferrard et al., 2016; Levie et al., 2018), and (2) *Matrix-valued operator* inspired by Fourier Neural Operator (FNO) (Li et al., 2021), where a matrix-valued $\mathbb{M}$ can be view as the convolutional kernel of a shift-invariant Transformers (Guibas et al., 2021). Details are given in Appendix B.

In this paper, we consider the matrix-valued form $\mathbb{M}$, since a powerful kernel with more parameters provides enough flexibility to adjust itself. Additionally, the experimental evidence in Section 6.3 shows the superiority of matrix-valued kernel against vector-valued one. The typical number of parameters in $\mathbb{M}$ is $N \times d \times d$, which can be significant, particularly for large-scale graphs with a large value of $N$. To efficiently model $\mathbb{M}$, we adopt parameter sharing among all frequency modes, employing only one Multi-Layer Perceptron (MLP). This results in a significant reduction of

Table 1: Comparison between classical graph convolution and WaveGC.

|  | Classical Graph Convolution | WaveGC |
|---|---|---|
| Formula | $\sigma(U\mathrm{diag}(\theta_\lambda)U^\top H \cdot W)$ | $\sigma\left([\Phi\mathbb{S}\circ\Phi H \,\|\, \Psi_s\mathbb{W}\circ\Psi_s H] \cdot W\right)$ |
| Kernel | $\mathrm{diag}(\theta_\lambda)$ (Vector) | $\mathbb{S}\,/\,\mathbb{W}$ (Matrix) |
| Bases | $U^\top$ (Fourier basis) | $\Phi\,/\,\Psi_s$ (Scaling / Wavelet basis) |

parameters in $\mathbb{M}$ from $N\times d\times d$ to $d\times d$. In this way, Eq. (3) becomes $\mathbb{M}*_{\mathcal{G}}\boldsymbol{X} = \mathcal{F}^{-1}\mathbb{M}\circ\mathcal{F}(\boldsymbol{X}) = \mathcal{F}^{-1}(\mathrm{MLP}(\mathcal{F}(\boldsymbol{X})))$.

For traditional point-wise vector-valued kernel, each frequency is independently scaled by one specific coefficient. Instead, our approach allows various frequency modes to interact, allowing them to collaboratively determine the optimal signal filtering strategy. Moreover, this sharing decreases the number of learnable parameters, and consequentially alleviates the risk of over-fitting caused by non-sharing, as in Section 6.3. An alternative method is presented in AFNO (Guibas et al., 2021), introducing a similar technique that offers improved efficiency but with a more intricate design.

### 3.3 WAVELET-BASED GRAPH CONVOLUTION

Until now, we have elaborated the proposed advancements on kernel and bases, and now discuss how to integrate these two aspects. Provided that we have $J$ wavelet $\{\Psi_{s_j}\}_{j=1}^{J}$ and one scaling function $\Phi$ constructed via the above Chebyshev decomposition, $\mathcal{F}:\mathbb{R}^{N\times d}\to\mathbb{R}^{N(J+1)\times d}$ in Eq. (3) is the stack of transforms from each component:

$$\mathcal{F}(\boldsymbol{H}^{(l)}) = \boldsymbol{T}\boldsymbol{H}^{(l)} = ((\Phi\boldsymbol{H}^{(l)})^\top\|(\Psi_{s_1}\boldsymbol{H}^{(l)})^\top\|...\|(\Psi_{s_J}\boldsymbol{H}^{(l)})^\top)^\top \in \mathbb{R}^{N(J+1)\times d}, \qquad (8)$$

where $\boldsymbol{T} = (\Phi^\top\|\Psi_{s_1}^\top\|...\|\Psi_{s_J}^\top)^\top$ is the overall transform and $\|$ means concatenation. Next, we check if the inverse $\mathcal{F}^{-1}$ exists. Considering $\boldsymbol{T}$ is not a square matrix, $\mathcal{F}^{-1}$ should be its pseudo-inverse as $(\boldsymbol{T}^\top\boldsymbol{T})^{-1}\boldsymbol{T}^\top$, where $\boldsymbol{T}^\top\boldsymbol{T} = \Phi\Phi^\top + \sum_{j=1}^{J}\Psi_{s_j}\Psi_{s_j}^\top = \boldsymbol{U}[h(\lambda)^2 + \sum_{j=1}^{J}g(s_j\lambda)^2]\boldsymbol{U}^\top$. Ideally, if $\boldsymbol{T}$ is imposed as **tight frames**, then $h(\lambda)^2 + \sum_{j=1}^{J}g(s_j\lambda)^2 = \boldsymbol{I}$ (Leonardi & Van De Ville, 2013), and $\boldsymbol{T}^\top\boldsymbol{T} = \boldsymbol{U}\boldsymbol{I}\boldsymbol{U}^\top = \boldsymbol{I}$. In this case, $\mathcal{F}^{-1} = (\boldsymbol{T}^\top\boldsymbol{T})^{-1}\boldsymbol{T}^\top = \boldsymbol{T}^\top$, and Eq. (3) becomes:

$$\boldsymbol{H}^{(l+1)} = \boldsymbol{T}^\top\mathbb{M}\circ\boldsymbol{T}\boldsymbol{H}^{(l)} = \Phi\mathbb{S}\circ\Phi\boldsymbol{H}^{(l)} + \sum_{j=1}^{J}\Psi_{s_j}\mathbb{W}_j\circ\Psi_{s_j}\boldsymbol{H}^{(l)} \in \mathbb{R}^{N\times d}, \qquad (9)$$

where we separate $\mathbb{M}$ into $\mathbb{S}$ and $\{\mathbb{W}\}_{j=0}^{J}$ as scaling kernel and different wavelet kernels.

**How to guarantee tight frames?** From above derivations, *tight frames* is a key for the simplification of inverse $\mathcal{F}^{-1}$ in Eq. (9). This can be guaranteed by $l_2$ norm on the above constructed wavelets and scaling function. For each eigenvalue $\lambda\in\boldsymbol{\Lambda}$, we have $v^2 = h(\lambda)^2 + \sum_{j=1}^{J}g(s_j\lambda)^2$, $\tilde{h}(\lambda) = h(\lambda)/v$, $\tilde{g}(s_j\lambda) = g(s_j\lambda)/v$. Then, $G(\boldsymbol{\Lambda}) = \tilde{h}(\boldsymbol{\Lambda})^2 + \sum_j\tilde{g}(s_j\boldsymbol{\Lambda})^2 = \boldsymbol{I}$ forms tight frames (Section 2). Thus, while the pseudo-inverse must theoretically exist, we can circumvent the necessity of explicitly calculating the pseudo-inverse.

Resembling the multi-head attention (Vaswani et al., 2017), we treat each wavelet transform as a "wavelet head", and concatenate them rather than sum them to get $\boldsymbol{H}^{(l+1)}\in\mathbb{R}^{N\times d}$:

$$\boldsymbol{H}^{(l+1)} = \sigma\left(\left[\Phi\mathbb{S}\circ\Phi\boldsymbol{H}^{(l)}\|\Psi_{s_1}\mathbb{W}_1\circ\Psi_{s_1}\boldsymbol{H}^{(l)}\|\ldots\|\Psi_{s_J}\mathbb{W}_J\circ\Psi_{s_J}\boldsymbol{H}^{(l)}\right]\cdot\boldsymbol{W}\right), \qquad (10)$$

where an outermost MLP increases the flexibility. Fig. 1 (b) presents the whole framework of our wavelet-based graph convolution, or WaveGC. For a better understanding, we compare classical graph convolution and WaveGC in Table. 1, where WaveGC contains only one wavelet for simplicity. Based on the differences shown in the table, WaveGC endows spectral graph convolution with the beneficial inductive bias of long-range dependency. This inductive bias supports the solid performance of the proposed model in most of the numerical experiments in section 6.

## 4 THEORETICAL PROPERTIES OF WAVEGC

Traditionally, wavelet is notable for its diverse receptive fields because of varying scales (Mallat, 1999). For graph wavelet, Hammond et al. (2011) were the first to prove the localization when scale $s \to 0$, but did not discuss the long-range case when $s \to \infty$. We further augment this discussion and demonstrate the effectiveness of the proposed WaveGC in capturing both short- and long-range information. Intuitively, a model's ability to integrate global information enables the reception and mixing of messages from distant nodes. Conversely, a model with a limited receptive field can only effectively mix local messages. Hence, assessing the degree of information 'mixing' becomes a key property. For this reason, we focus on the concept of *maximal mixing*:

**Definition 4.1. (Maximal mixing)** (Di Giovanni et al., 2023). *For a twice differentiable graph-function $y_G$ of node features $\boldsymbol{x}_i$, the maximal mixing induced by $y_G$ among the features $\boldsymbol{x}_a$ and $\boldsymbol{x}_b$ with nodes $a, b$ is*

$$\text{mix}_{y_G}(a, b) = \max_{\boldsymbol{x}_i} \max_{1 \leq \alpha, \beta \leq d} \left| \frac{\partial^2 y_G(\boldsymbol{X})}{\partial x_a^\alpha \partial x_b^\beta} \right|. \tag{11}$$

This definition is established in the context of graph-level task, and $y_G$ is the final output of an end-to-end framework, comprising the primary model and a readout function (e.g., mean, max) applied over the last layer. $\alpha$ and $\beta$ represent two entries of the $d$-dimensional features $\boldsymbol{x}_a$ and $\boldsymbol{x}_b$.

Next, we employ the concept of 'maximal mixing' on the WaveGC. For simplicity, we only take one wavelet basis $\Psi_s$ for analysis. The capacity of $\Psi_s$ on mixing information depends on two factors, i.e. $K$-order Chebyshev term and scale $s$. For a fair discussion on the effect of $s$ on message passing, we compare $\sigma(\Psi_s H W)$ and K-order message passing with the form of $\sigma(\sum_{j=0}^{K} \tau_j A^j H W)$, $\tau_j \in [0, 1]$:

**Theorem 4.2 (Short-range and long-range receptive fields).** *Given a large even number $K > 0$ and two random nodes $a$ and $b$, if the depths $m_\Psi$ and $m_A$ are necessary for $\sigma(\Psi_s H W)$ and $\sigma(\sum_{j=0}^{K} \tau_j A^j H W)$ to induce the same amount of mixing $\text{mix}_{y_G}(b, a)$, then the lower bounds of $m_\Psi$ and $m_A$, i.e. $L_{m_\Psi}$ and $L_{m_A}$, approximately satisfy the following relation when scale $s \to 0$:*

$$L_{m_\Psi} \approx \frac{P}{K} L_{m_A} + \frac{2|E|}{K\sqrt{d_a d_b}} \frac{mix_{y_G}(b, a)}{\gamma} \cdot \frac{1}{(\alpha^2 s^{2K})^{m_\Psi}}. \tag{12}$$

*Or, if $s \to \infty$, the relation becomes:*

$$L_{m_\Psi} \approx \frac{P}{K} L_{m_A} - \frac{2|E|}{K(K+1)^{2m_A} \tau_P{}^{2m_A} \sqrt{d_a d_b}} \frac{mix_{y_G}(b, a)}{\gamma}, \tag{13}$$

*where $P < K$ and $(\tau_P A^P)_{ba} = \max\{(\tau_m A^m)_{ba}\}_{m=0}^{K}$. $d_a$ and $d_b$ are degrees of two nodes, and $\alpha = \frac{C \cdot 2^K (K+1)}{K!}$. $\gamma = \sqrt{\frac{d_{max}}{d_{min}}}$, where $d_{max}/d_{min}$ is the maximum / minimum degree in the graph.*

The proof is provided in Appendix A.3. In Eq. (12), since the second term on the right-hand side is large ($s \to 0$), it required $\Psi_s$ to propagate more layers to mix the nodes. Conversely, if $s \to \infty$ (Eq. (13)), $\Psi_s$ will achieve the same degree of node mixing as $K$-hop message passing but with less propagation. Moreover, the greater the "mixing" $\text{mix}_{y_G}(b, a)$ is required between nodes, the fewer number of layers $L_{m_\Psi}$ is needed compared to $L_{m_A}$. To conclude, $\Psi_s$ presents the short- and long-range characteristics of WaveGC on message passing, while these characteristics do not derive from the order K of Chebyshev polynomials but from the scale $s$ exclusively.

## 5 WHY DO WE NEED DECOMPOSITION?

As shown in Fig. 1 (a), odd and even terms of Chebyshev polynomials meet the requirements on constructing wavelet after decomposition and transform. Additionally, each term is apt to be obtained according to the iteration formula, while infinite number of terms guarantee the expressiveness of the final composed wavelet. Next, we compare our decomposition solution with other related techniques:

• *Constructing wavelet via Chebyshev polynomials.* Both SGWT (Hammond et al., 2011) and GWNN (Xu et al., 2019a) firstly fix the shape of wavelets as cubic spline or exponential, followed by the approximation via Chebyshev polynomials. DEFT (Bastos et al., 2023) employs an MLP

or GNN network to learn the coefficients for Chebyshev bases. Comprehensively, the constructed wavelets from GWNN and DEFT fail to meet the wavelet admissible criteria because they impose no constrains to guarantee this point. SGWT fails to learn more flexible and expressive wavelet to better suit the dataset and task at hand.

• *Learnable graph bases.* If we uniformly learn the coefficients for all Chebyshev terms without decomposition, WaveGC degrades to a variant similar to ChebNet (Defferrard et al., 2016). However, mixture rather than decomposition blends the signals from different ranges, and the final spatial ranges cannot be precisely predicted and controlled. ChebNetII (He et al., 2022), reducing the Runge phenomenon via interpolation, confronts the same problem. Both JacobiConv (Wang & Zhang, 2022) and OptBasisGNN (Guo & Wei, 2023) emphasize the orthogonality of bases, but fail to manage multiple ranges information in spatial domain.

We provide numerical comparison and spectral visualization in section 6.2 for WaveGC against these related studies. In addition, although both WaveGC and Transformers can handle various node interaction ranges, the former adaptively learns which range must be emphasized, whereas the latter mixes short- and long-range information together without distinction. Thus, WaveGC surpasses Transformers in controlling distance based information.

# 6 NUMERICAL EXPERIMENTS

In this section, we evaluate the performance of WaveGC on both short-range and long-range benchmarks using the following datasets: (1) *Datasets for short-range tasks:* CS, Photo, Computer and CoraFull from the PyTorch Geometric (PyG) (Fey & Lenssen, 2019), and one large-size graph, i.e. ogbn-arxiv from Open Graph Benchmark (OGB) (Hu et al., 2020) (2) *Datasets for long-range tasks:* PascalVOC-SP (VOC), PCQM-Contact (PCQM), COCO-SP (COCO), Peptides-func (Pf) and Peptides-struct (Ps) from LRGB (Dwivedi et al., 2022). Please refer to Appendix D.2 for details of datasets.

Our aim is to mainly compare WaveGC and graph Transformers on capturing both short- and long-range information. Consequently, we replace the Transformer component in base models with WaveGC, while keeping the remaining components unchanged. The base models are Transformer (Vaswani et al., 2017), SAN (Kreuzer et al., 2021), and GraphGPS (Rampásek et al., 2022) because of the presence of the vanilla Transformer architecture in these three methods. Detailed descriptions of Transformer, SAN, and GraphGPS can be found in Appendix D.8. Additionally, we compare our method with other state-of-the-art models tailored to specific scenarios. Please refer to Appendix D.1 for implementation details.

## 6.1 BENCHMARKING WAVEGC

Table 2: Quantified results on short-range (S) and long-range (L) datasets compared to base models.

| Model | CS (S) | Photo (S) | Computer (S) | CoraFull (S) | ogbn-arxiv (S) | VOC (L) | PCQM (L) | COCO (L) | Pf (L) | Ps (L) |
|---|---|---|---|---|---|---|---|---|---|---|
| | Accuracy ↑ | Accuracy ↑ | Accuracy ↑ | Accuracy ↑ | Accuracy ↑ | F1 score ↑ | MRR ↑ | F1 score ↑ | AP ↑ | MAE ↓ |
| Transformer | 93.54±0.43 | 89.78±0.68 | 83.23±0.75 | 50.69±1.17 | 57.55±0.53 | 26.94±0.98 | 31.74±0.20 | **26.18±0.31** | 63.26±1.26 | 25.29±0.16 |
| **w/ WaveGC** | **94.77±0.30** | **93.93±0.62** | **89.69±0.66** | **63.97±1.49** | **71.69±0.26** | **29.42±0.94** | **33.30±0.10** | 25.11±0.25 | **65.18±0.77** | **25.04±0.26** |
| SAN | 93.82±0.41 | 94.92±0.38 | 91.31±0.33 | 64.15±1.01 | 70.25±0.26 | 32.30±0.39 | 33.50±0.03 | 25.92±1.58 | 63.84±1.21 | 26.83±0.43 |
| **w/ WaveGC** | **95.47±0.31** | **95.51±0.22** | **91.64±0.42** | **66.65±0.83** | **71.98±0.23** | **33.33±0.91** | **34.23±0.13** | **26.06±0.78** | **64.54±0.37** | **26.06±0.17** |
| GraphGPS | **95.47±0.31** | 94.47±0.46 | 89.51±0.74 | 62.79±0.72 | 71.45 ± 0.40 | 37.48±1.09 | 33.37±0.06 | 34.12±0.44 | 65.35±0.41 | 25.00±0.05 |
| **w/ WaveGC** | **95.89±0.34** | **95.37±0.44** | **91.00±0.48** | **69.14±0.78** | **72.85±0.24** | **40.24±0.28** | **34.50±0.02** | **35.01±0.22** | **70.10±0.27** | **24.95±0.07** |

For short-range (S) datasets, we follow the settings from (Chen et al., 2022b). For ogbn-arxiv, we use the public splits in OGB (Hu et al., 2020). For long-range datasets, we adhere to the experimental configurations outlined in (Dwivedi et al., 2022). (1) The results for comparing with base models are presented in Table 2. WaveGC consistently enhances the performance of base models across all datasets. (2) The results of the comparison with other SOTA models are shown in Table 3 and 4. Remarkably, our WaveGC demonstrates competitive performance and achieves the best results on CS, Photo, ogbn-arixv, VOC, COCO and Pf, as well as securing the second position on Computer and PCQM. In the experiments conducted on the five short-range datasets, the model is required to prioritize local information, while the five long-range datasets necessitate the handling of distant interactions. The results clearly demonstrate that the proposed WaveGC

Table 3: Qualified results on short-range tasks compared to baselines. **Bold**: Best, Underline: Runner-up, OOM: Out-of-memory, '*' Taken from original paper.

| Model | CS | Photo | Computer | CoraFull | ogbn-arxiv |
|---|---|---|---|---|---|
| | Accuracy ↑ | Accuracy ↑ | Accuracy ↑ | Accuracy ↑ | Accuracy ↑ |
| GCN (Kipf & Welling, 2017) | 92.92±0.12 | 92.70±0.20 | 89.65±0.52 | 61.76±0.14 | 71.74±0.29 |
| GAT (Velickovic et al., 2017) | 93.61±0.14 | 93.87±0.11 | 90.78±0.13 | 64.47±0.18 | 71.82±0.23 |
| APPNP (Gasteiger et al., 2018) | 94.49±0.07 | 94.32±0.14 | 90.18±0.17 | 65.16±0.28 | 71.90±0.25 |
| GPRGNN (Chien et al., 2020) | 95.13±0.09 | 94.49±0.14 | 89.32±0.29 | 67.12±0.31 | 71.78±0.18 |
| ChebNetII (He et al., 2022) | 95.39±0.39 | 94.71±0.25 | 89.85±0.85 | **72.18±0.58** | 72.32±0.23 |
| JacobiConv (Wang & Zhang, 2022) | 95.28±0.32 | 95.43±0.23* | 90.39±0.29* | 70.02±0.60 | 72.14±0.17 |
| OptBasisGNN (Guo & Wei, 2023) | 88.33±1.01 | 93.12±0.43 | 89.65±0.25 | 65.86±1.03 | 72.27±0.15* |
| Graphormer (Ying et al., 2021) | OOM | 92.74±0.14 | OOM | OOM | OOM |
| Nodeformer (Wu et al., 2022) | 95.28±0.28 | 95.27±0.22 | 91.12±0.43 | 61.82±0.81 | 59.90±0.42 |
| Specformer (Bo et al., 2023) | 93.43±0.35 | 95.48±0.32* | **92.19±0.48** | 68.41±0.65 | 72.37±0.18* |
| SGFormer (Wu et al., 2023b) | 93.63±0.36 | 94.08±0.35 | 91.17±0.38 | 69.66±0.63 | 72.63±0.13* |
| NAGphormer (Chen et al., 2022b) | 95.75±0.09* | 95.49±0.11* | 91.22±0.14* | 71.51±0.13* | 71.79±0.37 |
| Exphormer (Shirzad et al., 2023) | 95.77±0.15* | 95.27±0.42* | 91.59±0.31* | 65.42±0.75 | 72.44±0.28* |
| **WaveGC (ours)** | **95.89±0.34** | **95.51±0.22** | 91.64±0.42 | 69.14±0.78 | **72.85±0.24** |

Table 4: Qualified results on long-range tasks compared to baselines. **Bold**: Best, Underline: Runner-up, OOM: Out-of-memory, '†' Original code run by us.

| Model | VOC | PCQM | COCO | Pf | Ps |
|---|---|---|---|---|---|
| | F1 score ↑ | MRR ↑ | F1 score ↑ | AP ↑ | MAE ↓ |
| GCN (Kipf & Welling, 2017) | 12.68±0.60 | 32.34±0.06 | 08.41±0.10 | 59.30±0.23 | 34.96±0.13 |
| GINE (Xu et al., 2019b) | 12.65±0.76 | 31.80±0.27 | 13.39±0.44 | 54.98±0.79 | 35.47±0.45 |
| GatedGCN (Bresson & Laurent, 2017) | 28.73±2.19 | 32.18±0.11 | 26.41±0.45 | 58.64±0.77 | 34.20±0.13 |
| ChebNetII$^\dagger$ ($Heet\ al.$, 2022) | 36.45±0.52 | 34.34±0.10 | 26.02±0.53 | 68.19±0.27 | 26.18±0.58 |
| JacobiConv$^\dagger$ ($Wang\&Zhang$, 2022) | 32.52±0.87 | 34.24±0.24 | 30.46±0.46 | 68.00±0.53 | 25.20±0.21 |
| OptBasisGNN$^\dagger$ ($Guo\&Wei$, 2023) | 33.83±0.61 | 32.42±0.45 | 22.02±0.18 | 61.92±0.75 | 25.61±0.19 |
| GraphViT (He et al., 2023) | 30.46±1.15$^\dagger$ | 32.80±0.05$^\dagger$ | 27.29±0.49$^\dagger$ | 69.42±0.75 | **24.49±0.16** |
| GRIT (Ma et al., 2023) | OOM$^\dagger$ | 34.33±0.26$^\dagger$ | OOM$^\dagger$ | 69.88±0.82 | 24.60±0.12 |
| Specformer (Bo et al., 2023)$^\dagger$ | 35.64±0.85 | 33.73±0.27 | 25.40±0.55 | 66.86±0.64 | 25.50±0.14 |
| Exphormer (Shirzad et al., 2023) | 39.60±0.27 | **36.37±0.20** | 34.30±0.08 | 65.27±0.43 | 24.81±0.07 |
| **WaveGC (ours)** | **40.24±0.28** | 34.50±0.02 | **35.01±0.22** | **70.10±0.27** | 24.95±0.07 |

consistently outperforms traditional graph convolutions and GTs in effectively aggregating both local and long-range information.

## 6.2 EFFECTIVENESS OF GENERAL WAVELET BASES

Table 5: Numerical comparison between WaveGC and other graph wavelets and polynomial bases

| Model | Graph wavelet | | | Polynomial bases | | | | **Ours** |
|---|---|---|---|---|---|---|---|---|
| | SGWT | DEFT | GWNN | ChebNet* | ChebNetII | JacobiConv | OptBasisGNN | **WaveGC** |
| Computer (Accuarcy ↑) | 89.05 | 91.34 | 90.36 | 90.28 | 89.85 | 90.39 | 89.65 | **91.64** |
| VOC (F1 score ↑) | 31.22 | 35.98 | 25.60 | 37.80 | 36.45 | 32.52 | 33.83 | **40.24** |

In this section, we compare the learnt wavelet bases from WaveGC with other baselines, including three graph wavelets (i.e. SGWT (Hammond et al., 2011), DEFT (Bastos et al., 2023), GWNN (Xu et al., 2019a)) and four polynomial bases as in Table 5. Here, ChebNet* is a variant of our WaveGC where the only change is to combine odd and even terms without decomposition. Therefore, the improvement of WaveGC over ChebNet* reflects the effectiveness of decoupling operation. The numerical comparison on `Computer` and `PascalVOC-SP` is shown in Table. 5, which demonstrates obvious gains from WaveGC especially on long-range `PascalVOC-SP`.

To address the performance gap observed on the VOC dataset, we provide insights through the spectral visualization of various bases in Fig. 2 [1]. These visualizations again confirm the disadvantages of other bases analyzed in section 5. For our WaveGC, the figure intuitively demonstrates that the unit wavelet got by decomposition of Chebyshev polynomials strictly meets the admissibility criteria, as Eq. equation 1, while the corresponding base scaling function supplements the direct current

---

[1] We do not visualize OptBasisGNN, as it learns bases with implicit recurring relation.

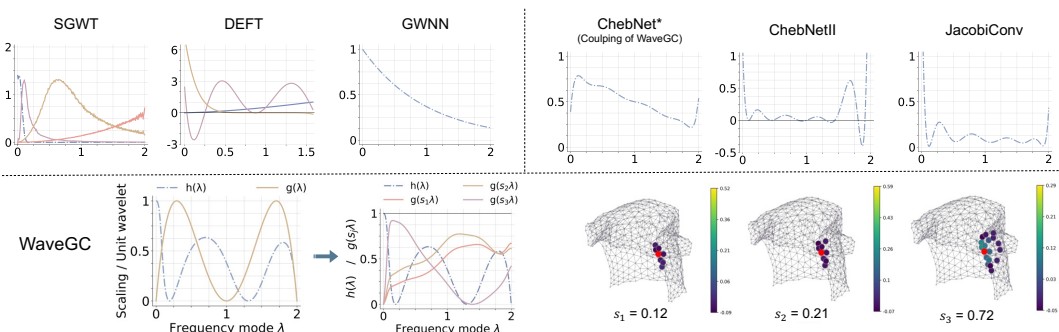

Figure 2: The spectral and spatial visualization of different bases on `PascalVOC-SP`.

signals at $\lambda = 0$. We also depicts the signal distribution over the topology centered on the target node (the red-filled circle). The receptive field of the central node expands with the increasing of scale $s$, aggregating both short- and long-range information simultaneously. More analyses are given in Appendix D.3.

### 6.3 EFFECTIVENESS OF MATRIX-VALUED KERNEL

Newly proposed matrix-valued kernel and weight-sharing technique constitutes the first advancement over graph convolution. In this section, we deeply explore the effectiveness of these two designs.

Table 6: Comparison between *Matrix-valued* and *Vector-valued* kernels.

| Kernel | Photo (Accuracy ↑) | Ps (MAE ↓) |
|---|---|---|
| *Vector-valued* | 94.61 | 25.30 |
| ***Matrix-valued*** | **95.37** | **24.95** |

Table 7: Comparison between *sharing* and *non-sharing* kernel weights

| Result (Parameters) | CoraFull (Accuracy ↑) | Ps (MAE ↓) |
|---|---|---|
| *Non-sharing* | 67.67 (883,215) | 26.22 (1,410,029) |
| ***Sharing*** | **69.14 (621,135)** | **24.95 (534,701)** |

Observing the results from Table 6, the better performance from matrix-valued kernel indicates that more parameters on parameterizing kernels lead to enhanced feature learning. Meanwhile, upon analysis of Table 7, specifically learning kernels for each frequency does not yield improvement, and may even degrade performance. This degradation may be attributed to the large number of involved parameters, with potentially over-fitting. Matrix-valued kernels necessitates a mapping from each eigenvalue embedding to a specific matrix, $f : \mathbb{R}^d \to \mathbb{R}^{d \times d}$, which involves a MLP with weight dimension $\mathbb{R}^{d \times d \times d}$, $d$ is the embedding dimension. This results in a significant increase in the number of learnable parameters, as seen with $d = 96$ in Ps, where the total number is nearly $96 \times 96 \times 96 = 884,736$.

**Other experiments** In Appendix D.4, we analyze the effect of different components, explore the mixing benefit between different architectures WaveGC, GCN and Transformer, and test differences between WaveGC and ChebNet. In Appendix D.5, we conduct a deep study on time and space consumption. Moreover, we also test our WaveGC on heterophily scenarios in Appendix D.6. Though WaveGC does not initially target on this topic, its positive performances convince us its potential to be further explored. In the end, we test the sensitivity of two important hyper-parameters in Appendix D.7.

## 7 CONCLUSION

In this study, we proposed a novel graph convolution operation based on wavelets (WaveGC), establishing its theoretical capability to capture information at both short and long ranges through a multi-resolution approach.

**Limitation.** One potential limitation of WaveGC is the computational complexity. The main contribution of WaveGC is to address long-range interactions in graph convolution, so it inevitably

establishes connections between most of nodes. This results in the same $O(N^2)$ complexity as Transformer (Vaswani et al., 2017) and Specformer (Bo et al., 2023). A possible solution is to decrease the number of considered frequency modes from $N$ to $\nu$. In this way, the complexity is reduced to $O(\nu \cdot N)$. This operation makes WaveGC run on large-scale graph, i.e. `ogbn-arxiv`, and the good performance supports this simplification. Moreover, the eigen-decomposition process involves $O(N^3)$ complexity. However, this decomposition is performed only once, prior to all training experiments. To accelerate the decomposition, we may adopt randomized SVD Halko et al. (2009) with complexity $O(N^2 \log K)$. Future work will focus on further simplification and scaling up to larger graphs.

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

## A  THEORETICAL PROOF

Firstly, we give two auxiliary but indispensable lemma and theorem. Let starts from the formula $\sigma(\Psi_s HW)$. In this equation, we bound the first derivate of non-linear function as $|\sigma'| < c_\sigma$, and set $||W|| \leq w$, where $|| \cdot ||$ is the operator norm. First, we give an upper bound for each entry in $\Psi_s$.

**Lemma A.1** (**Upper bound for graph wavelet**). *Let $\Psi = \boldsymbol{U} g(\Lambda) \boldsymbol{U}^T$. Given a large even number $K > 0$, then for $\forall i, j \in V \times V$, we have:*

$$(\Psi_s)_{ij} < \left( \alpha(\hat{\boldsymbol{A}})^{K/2} s^K \right)_{ij}, \quad \alpha = \frac{C \cdot 2^K (K+1)}{K!}. \tag{14}$$

The proof is given in Appendix A.1. In this lemma, we assume $g$ is smooth enough at $\lambda = 0$. For fair comparison with traditional K-hop message passing framework $\sigma(\sum_{j=0}^{K} \tau_j A^j HW)$, we just test the flexibility with the similar form $\sigma(\Psi_s HW)$. In this case, we derive the depth $m_\Psi$ necessary for this wavelet basis $\Psi_s$ to induce the amount of mixing $\text{mix}_{y_G}(a, b)$ between two nodes $a$ and $b$.

**Theorem A.2** (**The least depth for mixing**). *Given commute time $\tau(a, b)$ (Lovász, 1993) and number of edges $|E|$. If $\Psi_s$ generates mixing $\text{mix}_{y_G}(b, a)$, then the number of layers $m_\Psi$ satisfies*

$$m_\Psi \geq \frac{\tau(a, b)}{2K} + \frac{2|E|}{K\sqrt{d_a d_b}} \left[ \frac{\text{mix}_{y_G}(b, a)}{\gamma(\alpha^2 s^{2K})^{m_\Psi}} - \frac{1}{\lambda_1}(\gamma + |1 - \lambda^*|^{K m_\Psi + 1}) \right], \tag{15}$$

*where $d_a$ and $d_b$ are degrees of two nodes, $\gamma = \sqrt{\frac{d_{max}}{d_{min}}}$, and $|1 - \lambda^*| = \max_{0 < n \leq N-1} |1 - \lambda_n| < 1$.*

The proof is given in Appendix A.2. In the following subsections, we firstly prove these lemma and theorem, and finally give the complete proof of Theorem 4.2.

### A.1  PROOF OF LEMMA A.1 (UPPER BOUND FOR GRAPH WAVELET)

*Proof.* We aim to investigate the properties of filters $\Psi_{s_j} = \boldsymbol{U} g(s_j \lambda) \boldsymbol{U}^\top$ to capture both global and local information, corresponding to the cases $s_j \to 0$ and $s_j \to \infty$, respectively. In the former case, as $s_j$ approaches zero, $g(s_j \lambda)$ tends towards $g(0)$. For the latter case, the spectral information becomes densely distributed and concentrated near zero. Hence, the meaningful analysis of $g(\lambda)$ primarily revolves around $\lambda = 0$. Expanding $g(\lambda)$ using Taylor's series around $\lambda = 0$, we get:

$$g(\lambda) = \sum_{k=0}^{K} C_k \frac{\lambda^k}{k!} + g^{(K+1)}(\lambda^*) \frac{\lambda^{K+1}}{(K+1)!} \approx \sum_{k=0}^{K} C_k \frac{\lambda^k}{k!}, \tag{16}$$

where we neglect the high-order remainder term. Next, we have

$$(\Psi)_{ij} = \left( \boldsymbol{U} g(\Lambda) \boldsymbol{U}^T \right)_{ij} = \left( \sum_{k=0}^{K} C_k \frac{\hat{\mathcal{L}}^k}{k!} \right)_{ij}$$

$$= \left( \sum_{k=0}^{K} \frac{C_k}{k!} (\boldsymbol{I} - \hat{\boldsymbol{A}})^k \right)_{ij} = \left( \sum_{k=0}^{K} \frac{C_k}{k!} \sum_{p=0}^{k} \binom{k}{p} (-\hat{\boldsymbol{A}})^p \right)_{ij}$$

$$< \left( \sum_{k=0}^{K} \frac{C_k}{k!} \sum_{p=0}^{k} \binom{k}{p} (\hat{\boldsymbol{A}})^p \right)_{ij} = \left( \sum_{k=0}^{K} \frac{C_k}{k!} \sum_{p=0}^{k} \frac{k!}{(k-p)!p!} (\hat{\boldsymbol{A}})^p \right)_{ij}$$

$$= \left( \sum_{k=0}^{K} C_k \sum_{p=0}^{k} \frac{(\hat{\boldsymbol{A}})^p}{(k-p)!p!} \right)_{ij}. \tag{17a}$$

We introduce a new parameter $\mu = \frac{\left( \sum_{k=0}^{K-1} C_k \sum_{p=0}^{k} \frac{(\hat{\boldsymbol{A}})^p}{(k-p)!p!} \right)_{ij}}{\left( C_K \sum_{p=0}^{K} \frac{(\hat{\boldsymbol{A}})^p}{(K-p)!p!} \right)_{ij}}$, so the above relation becomes:

$$(\Psi)_{ij} < \left( (\mu + 1) C_K \sum_{p=0}^{K} \frac{(\hat{\boldsymbol{A}})^p}{(K-p)!p!} \right)_{ij} = \left( C \sum_{p=0}^{K} \frac{(\hat{\boldsymbol{A}})^p}{(K-p)!p!} \right)_{ij}, \tag{18}$$

where we set $C = (\mu + 1)C_K$. Then, let us explore the expression $\epsilon_{ij}^p = \frac{(\hat{\boldsymbol{A}})_{ij}^p}{(K-p)!p!}$. First, we will address the denominator $(K - p)!p!$. As $p$ increases, this denominator experiences a sharp decrease followed by a rapid increase. The minimum value occurs at $(K/2)!(K/2)!$ when $p = K/2$, assuming $K$ is even. Second, let's analyze the numerator $(\hat{\boldsymbol{A}})_{ij}^p$, which involves repeated multiplication of $\hat{\boldsymbol{A}}$. According to Theorem 1 in (Li et al., 2018), this repeated multiplication causes $(\hat{\boldsymbol{A}})^p$ to converge to the eigenspaces spanned by the eigenvector $D^{-1/2}\mathbf{1}$ of $\lambda = 0$, where $\mathbf{1} = (1, 1, \ldots, 1) \in \mathbb{R}^{n\;2}$. Then, let us assume there exists a value $p^*$ beyond which the change in $(\hat{\boldsymbol{A}})^p$ becomes negligible. Given that $K$ is a large even number, we can infer that $K/2 \gg p^*$. Thus, when $(K - p)!p!$ sharply decreases, $(\hat{\boldsymbol{A}})^p$ has already approached a stationary state. Consequently, $\max \epsilon_{ij}^p = \frac{(\hat{\boldsymbol{A}})_{ij}^{K/2}}{(K/2)!(K/2)!}$, where the denominator reaches its minimum. Thus, we have

$$(\Psi)_{ij} < \left( C \sum_{p=0}^{K} \frac{(\hat{\boldsymbol{A}})^p}{(K-p)!p!} \right)_{ij}$$

$$< C(K + 1) \left( \frac{(\hat{\boldsymbol{A}})^{K/2}}{(K/2)!(K/2)!} \right)_{ij}$$

$$< \left( \frac{C \cdot 2^K (K + 1)}{K!} (\hat{\boldsymbol{A}})^{K/2} \right)_{ij}. \tag{19a}$$

We have $\frac{1}{(K/2)!(K/2)!} < \frac{2^K}{K!}$ given that

$$(K/2)!(K/2)! = (\frac{K}{2} \cdot \frac{K-2}{2} \cdots \frac{4}{2} \cdot \frac{2}{2})(\frac{K}{2} \cdot \frac{K-2}{2} \cdots \frac{4}{2} \cdot \frac{2}{2})$$

$$> (\frac{K}{2} \cdot \frac{K-2}{2} \cdots \frac{4}{2} \cdot \frac{2}{2})(\frac{K-1}{2} \cdot \frac{K-3}{2} \cdots \frac{3}{2} \cdot \frac{1}{2}) \tag{20}$$

$$= \underbrace{\frac{K \cdot K-1 \cdot K-2 \cdot K-3 \ldots 4 \cdot 3 \cdot 2 \cdot 1}{2 \cdot 2 \cdot 2 \cdot 2 \ldots 2 \cdot 2 \cdot 2 \cdot 2}}_{K \text{ terms}} = \frac{K!}{2^K}.$$

With $\alpha = \frac{C \cdot 2^K (K+1)}{K!}$ and scale $s$, Eq. (19a) can be finally written as

$$(\Psi_s)_{ij} < \left( \alpha (\hat{\boldsymbol{A}})^{K/2} s^K \right)_{ij}. \tag{21}$$

$\square$

## A.2 PROOF OF THEOREM A.2 (THE LEAST DEPTH FOR MIXING)

For this section, we mainly refer to the proof from (Di Giovanni et al., 2023).

**Preliminary.** For simplicity, we follow (Di Giovanni et al., 2023) to denote some operations utilized in this section. As stated, we consider the message passing formula $\sigma(\Psi_s HW)$. First, we denote $\boldsymbol{h}_a^{(l),\alpha}$ as the $\alpha$-th entry of the embedding $\boldsymbol{h}_a^{(l)}$ for node $a$ at the $l$-th layer. Then, we rewrite the formula as:

$$\boldsymbol{h}_a^{(l),\alpha} = \sigma(\widetilde{\boldsymbol{h}}_a^{(l-1),\alpha}), \quad 1 \le \alpha \le d, \tag{22}$$

where $\widetilde{\boldsymbol{h}}_a^{(l-1),\alpha} = (\Psi_s HW)_a$ is the entry $\alpha$ of the pre-activated embedding of node $a$ at layer $l$. Given nodes $a$ and $b$, we denote the following differentiation operations:

$$\nabla_a \boldsymbol{h}_b^{(l)} := \frac{\partial \boldsymbol{h}_b^{(l)}}{\partial \boldsymbol{x}_a}, \quad \nabla_{ab}^2 \boldsymbol{h}_i^{(l)} := \frac{\partial^2 \boldsymbol{h}_i^{(l)}}{\partial \boldsymbol{x}_a \partial \boldsymbol{x}_b}. \tag{23}$$

Next, we firstly derive upper bounds on $\nabla_a \boldsymbol{h}_b^{(l)}$, and then on $\nabla_{ab}^2 \boldsymbol{h}_i^{(l)}$.

---

[2]Simple proof. $(\hat{\boldsymbol{A}})^p = \boldsymbol{U}(\boldsymbol{I} - \boldsymbol{\Lambda})^p \boldsymbol{U}^\top = \sum_{i=0}^n (1 - \lambda_i)^p \boldsymbol{u_1}\boldsymbol{u_1}^\top$. Provided only $1 - \lambda_0 = 1$ and $1 - \lambda_i \in (-1, 1)$ for other eigenvalues, with $p \to \infty$, only $(1 - \lambda_0)^p = 1$ but $(1 - \lambda_i)^p \to 0$. Thus, we have $(\hat{\boldsymbol{A}})^p \to \boldsymbol{u_1}\boldsymbol{u_1}^\top$, where $\boldsymbol{u_1} = D^{-1/2}\mathbf{1}$

**Lemma A.3.** *Given the message passing formula $\sigma(\Psi_s HW)$, let assume $|\sigma'| \leq c_\sigma$ and $||W|| \leq w$, where $|| \cdot ||$ is the operator norm. For two nodes $a$ and $b$ after $l$ layers of message passing, the following holds:*

$$||\nabla_a \boldsymbol{h}_b^{(l)}|| \leq (c_\sigma w)^l (\boldsymbol{B}^l)_{ba}, \tag{24}$$

*where $\boldsymbol{B}_{ba} = \left( \alpha(\hat{\boldsymbol{A}})^{K/2} s^K \right)_{ba}$.*

*Proof.* If $l = 1$ and we fix entries $1 \leq \alpha, \beta \leq d$, then we have:

$$(\nabla_a \boldsymbol{h}_b^{(1)})_{\alpha\beta} = (\text{diag}(\sigma'(\widetilde{\boldsymbol{h}}_b^{(0)}))(\boldsymbol{W}^{(1)}\Psi_{ba}\boldsymbol{I}))_{\alpha\beta}. \tag{25}$$

With Cauchy–Schwarz inequality, we bound the left hand side by

$$||\nabla_a \boldsymbol{h}_b^{(1)}|| \leq ||\text{diag}(\sigma'(\widetilde{\boldsymbol{h}}_b^{(0)}))|| \cdot ||\boldsymbol{W}^{(1)}\Psi_{ba}||$$
$$\leq c_\sigma w \boldsymbol{B}_{ba}.$$

Next, we turn to a general case where $l > 1$:

$$(\nabla_a \boldsymbol{h}_b^{(l)})_{\alpha\beta} = (\text{diag}(\sigma'(\widetilde{\boldsymbol{h}}_b^{(l-1)})(W \sum_j \Psi_{bj} \nabla_a \boldsymbol{h}_j^{(m-1)}))_{\alpha\beta}. \tag{27}$$

Then, we can use the induction step to bound the above equation:

$$||\nabla_a \boldsymbol{h}_b^{(l)}|| \leq (c_\sigma w)^l | \sum_{j_0} \sum_{j_1} \cdots \sum_{j_{l-2}} \Psi_{bj_0} \Psi_{j_0 j_1} \ldots \Psi_{j_{l-3}j_{l-2}} \Psi_{j_{l-2}a}|$$
$$\leq (c_\sigma w)^l (\boldsymbol{B}^l)_{ba}. \tag{28}$$

In Eq. (28), we implicitly use $|\Psi_s^l|_{ba} < \left( \alpha(\hat{\boldsymbol{A}})^{K/2} s^K \right)_{ba}^l = \boldsymbol{B}_{ba}^l$. Similar to proof given in Appendix A.1, we can give the following proof:

$$|\Psi_s^l|_{ba} = \left| \boldsymbol{U} g(s\Lambda)^l \boldsymbol{U}^T \right|_{ba} = \left| s^{lK} C^l \frac{\hat{\mathcal{L}}^{lK}}{K!^l} \right|_{ba}$$

$$= \left| s^{lK} \frac{C^l}{K!^l} (\boldsymbol{I} - \hat{\boldsymbol{A}})^{lK} \right|_{ba} = \left| s^{lK} \frac{C^l}{K!^l} \sum_{p=0}^{lK} \binom{lK}{p} (-\hat{\boldsymbol{A}})^p \right|_{ba}$$

$$< \left( s^{lK} \frac{C^l}{K!^l} \sum_{p=0}^{lK} \binom{lK}{p} (\hat{\boldsymbol{A}})^p \right)_{ba} = \left( s^{lK} \frac{C^l}{K!^l} \sum_{p=0}^{lK} \frac{(lK)!}{(lK-p)!p!} (\hat{\boldsymbol{A}})^p \right)_{ba}$$

$$= \left( s^{lK} \frac{C^l(lK)!}{K!^l} \sum_{p=0}^{lK} \frac{(\hat{\boldsymbol{A}})^p}{(lK-p)!p!} \right)_{ba} < \left( s^{lK} \frac{C^l(lK)!}{K!^l} (lK+1) \left( \frac{(\hat{\boldsymbol{A}})^{lK/2}}{(lK/2)!(lK/2)!} \right) \right)_{ba}$$

$$< \left( s^{lK} \frac{C^l(lK)!}{K!^l} (lK+1) \frac{2^{lK}}{(lK)!} (\hat{\boldsymbol{A}})^{lK/2} \right)_{ba} = \left( s^{lK} \frac{C^l \cdot 2^{lK}(lK+1)}{K!^l} (\hat{\boldsymbol{A}})^{lK/2} \right)_{ba}$$

$$< \left( s^{lK} \frac{C^l \cdot 2^{lK}(K+1)^l}{K!^l} (\hat{\boldsymbol{A}})^{lK/2} \right)_{ba} = \left( \alpha(\hat{\boldsymbol{A}})^{K/2} s^K \right)_{ba}^l, \tag{29}$$

where in the last line, we utilize the relation $lK + 1 < (K+1)^l$. $\qquad\square$

**Lemma A.4.** *Given the message passing formula $\sigma(\Psi_s HW)$, let assume $|\sigma'|, |\sigma''| \leq c_\sigma$ and $||W|| \leq w$, where $|| \cdot ||$ is operator norm. For nodes $i$, $a$ and $b$ after $l$ layers of message passing, the following holds:*

$$||\nabla_{ab}^2 \boldsymbol{h}_i^{(l)}|| \leq \sum_{k=0}^{l-1} \sum_{j \in V} (c_\sigma w)^{2l-k-1} w (\boldsymbol{B}^{l-k})_{jb} (\boldsymbol{B}^k)_{ij} (\boldsymbol{B}^{l-k})_{ja}, \tag{30}$$

*where $\boldsymbol{B}_{ba} = \left( \alpha(\hat{\boldsymbol{A}})^{K/2} s^K \right)_{ba}$.*

*Proof.* Considering $\nabla_{ab}^2 \boldsymbol{h}_i^{(l)} \in \mathbb{R}^{d \times (d \times d)}$, we refer to (Di Giovanni et al., 2023) to use the following ordering for indexing the columns:

$$\frac{\partial^2 \boldsymbol{h}_i^{(l),\alpha}}{\partial x_b^\beta \partial x_a^\gamma} := (\nabla_{ab}^2 \boldsymbol{h}_i^{(l)})_{\alpha,d(\beta-1)+\gamma}. \tag{31}$$

Similar to the proof of Lemma A.3, we firstly focus on $m = 1$:

$$(\nabla_{ab}^2 \boldsymbol{h}_i^{(1)})_{\alpha,d(\beta-1)+\gamma} = (\text{diag}(\sigma''(\widetilde{\boldsymbol{h}}_i^{(0),\alpha}))(\boldsymbol{W}^{(1)} \Psi_{ib} \boldsymbol{I})_{\alpha\gamma} \times (\boldsymbol{W}^{(1)} \Psi_{ia} \boldsymbol{I})_{\alpha\beta}. \tag{32}$$

We bound the left-hand side as:

$$||\nabla_{ab}^2 \boldsymbol{h}_i^{(1)}|| \leq (c_\sigma w)(w|\boldsymbol{B}_{ib}||\boldsymbol{B}_{ia}|). \tag{33}$$

Then, for $m > 1$:

$$\begin{aligned}
&(\nabla_{ab}^2 \boldsymbol{h}_i^{(l)})_{\alpha,d(\beta-1)+\gamma} \\
&= \underbrace{\text{diag}(\sigma''(\widetilde{\boldsymbol{h}}_i^{(l-1),\alpha})(W \sum_j \Psi_{ij} \nabla_a \boldsymbol{h}_j^{(l-1)}) \times (W \sum_j \Psi_{ij} \nabla_b \boldsymbol{h}_j^{(l-1)})}_{\boldsymbol{R}} \\
&+ \underbrace{\text{diag}(\sigma'(\widetilde{\boldsymbol{h}}_i^{(l-1),\alpha})(\boldsymbol{W}^{(m)} \sum_j \Psi_{ij} \nabla_{ab}^2 \boldsymbol{h}_j^{(l-1)})}_{\boldsymbol{Z}}.
\end{aligned} \tag{34}$$

We denote $||\nabla_j \boldsymbol{h}_i^{(l-1)}||$ as $(D\boldsymbol{h}^{(l-1)})_{ij}$, and $||\nabla_{ab}^2 \boldsymbol{h}_i^{(l-1)}||$ as $(D^2 \boldsymbol{h}^{(l-1)}{}_{ba})_i$. To bound $\boldsymbol{R}$, we deduce as follows:

$$\begin{aligned}
||\boldsymbol{R}|| &\leq c_\sigma (w \sum_j \boldsymbol{B}_{ij} ||\nabla_a \boldsymbol{h}_j^{(l-1)}||) \times (w \sum_j \boldsymbol{B}_{ij} ||\nabla_b \boldsymbol{h}_j^{(l-1)}||) \\
&= c_\sigma w (w \boldsymbol{B} D \boldsymbol{h}^{(l-1)})_{ib} (\boldsymbol{B} D \boldsymbol{h}^{(l-1)})_{ia} \\
&\leq c_\sigma w (w \boldsymbol{B}(c_\sigma w)^{l-1} \boldsymbol{B}^{l-1})_{ib} (\boldsymbol{B}(c_\sigma w)^{l-1} \boldsymbol{B}^{l-1})_{ia} \\
&= (c_\sigma w)^{2l-1} (w(\boldsymbol{B}^l)_{ib} (\boldsymbol{B}^l)_{ia}),
\end{aligned} \tag{35a}$$

where we utilize the conclusion from Theorem A.3 in (35a). For term $\boldsymbol{Z}$, we have:

$$\begin{aligned}
||\boldsymbol{Z}|| &\leq c_\sigma w (\boldsymbol{B} D^2 \boldsymbol{h}^{(l-1)})_i \\
&\leq c_\sigma w \sum_s \boldsymbol{B}_{is} \sum_{k=0}^{l-2} \sum_{j \in V} (c_\sigma w)^{2l-2-k-1} w (\boldsymbol{B}^{l-1-k})_{jb} (\boldsymbol{B}^k)_{sj} (\boldsymbol{B}^{l-1-k})_{ja} \\
&= \sum_{k=0}^{l-2} \sum_{j \in V} (c_\sigma w)^{2l-2-k} (\boldsymbol{B}^{l-1-k})_{jb} (\boldsymbol{B}^{k+1})_{ij} (\boldsymbol{B}^{l-1-k})_{ja} \\
&= \sum_{k=1}^{l-1} \sum_{j \in V} (c_\sigma w)^{2l-1-k} (\boldsymbol{B}^{l-k})_{jb} (\boldsymbol{B}^k)_{ij} (\boldsymbol{B}^{l-k})_{ja},
\end{aligned} \tag{36a}$$

where in (36a), we recursively use the Eq. (34). Finally, we finish the proof as:

$$\begin{aligned}
||\nabla_{ab}^2 \boldsymbol{h}_i^{(l)}|| &\leq ||\boldsymbol{R}|| + ||\boldsymbol{Z}|| \\
&\leq \sum_{k=0}^{l-1} \sum_{j \in V} (c_\sigma w)^{2l-1-k} (\boldsymbol{B}^{l-k})_{jb} (\boldsymbol{B}^k)_{ij} (\boldsymbol{B}^{l-k})_{ja}.
\end{aligned} \tag{37}$$

$\square$

With Lemma A.3 and A.4, now we give the following theorem.

**Theorem A.5.** *Consider the message passing formula $\sigma(\Psi_s HW)$ with $m_\Psi$ layers, the induced mixing $mix_{y_G}(b, a)$ over the features of nodes $a$ and $b$ satisfies:*

$$mix_{y_G}(b, a) \le \sum_{l=0}^{m_\Psi - 1} (c_\sigma w)^{(2m_\Psi - l - 1)} \left( w \left( \boldsymbol{B}^{m_\Psi - l} \right)^\top diag \left( \boldsymbol{1}^\top \boldsymbol{B}^l \right) \boldsymbol{B}^{m_\Psi - l} \right)_{ab}, \qquad (38)$$

*where $\boldsymbol{B}_{ba} = \left( \alpha(\hat{\boldsymbol{A}})^{K/2} s^K \right)_{ba}$ and $\boldsymbol{1} \in \mathbb{R}^n$ is the vector of ones.*

*Proof.* Here, we define the prediction function $y_G : N \times d \to d$ on $G$ as $y_G^{(m_\Psi)} = \texttt{Readout}(\boldsymbol{H}^{(m_\Psi)}\boldsymbol{\theta})$, where `Readout` is to gather all nodes embeddings to get the final graph embedding, $\boldsymbol{H}^{(m_\Psi)}$ is the node embedding matrix after $m_\Psi$ layers and $\boldsymbol{\theta}$ is the learnable weight for graph-level task. If we set `Readout = sum`, we derive:

$$\text{mix}_{y_G}(b, a) = \max_x \max_{1 \le \beta, \gamma \le d} \left| \frac{\partial^2 y_G^{(m_\Psi)}(\boldsymbol{X})}{\partial \boldsymbol{x}_a^\beta \partial \boldsymbol{x}_b^\gamma} \right|$$

$$\le \sum_{i \in V} \left| \sum_{\alpha=1}^d \theta_\alpha \frac{\partial^2 h_i^{(m_\Psi), \alpha}}{\partial \boldsymbol{x}_a^\beta \partial \boldsymbol{x}_b^\gamma} \right|$$

$$= \sum_{i \in V} ||(\nabla_{ab}^2 \boldsymbol{h}_i^{(m_\Psi)})^\top \boldsymbol{\theta}||$$

$$\le \sum_{i \in V} ||\nabla_{ab}^2 \boldsymbol{h}_i^{(m_\Psi)}|| \tag{39a}$$

$$\le \sum_{k=0}^{m_\Psi - 1} (c_\sigma w)^{(2m_\Psi - k - 1)} \left( w \left( \boldsymbol{B}^{m_\Psi - k} \right)^\top diag \left( \boldsymbol{1}^\top \boldsymbol{B}^k \right) \boldsymbol{B}^{m_\Psi - k} \right)_{ab}, \tag{39b}$$

$$\square$$

where in (39a), we assume the norm $||\boldsymbol{\theta}|| \le 1$. In (39b), we use the results from Lemma A.4. This upper bound still holds if `Readout` is chosen as `MEAN` or `MAX` (Di Giovanni et al., 2023).

In theorem A.5, we can assume that $c_\sigma$ to be smaller or equal than one, which is satisfied by the majority of current active functions. Furthermore, considering the normalization (e.g., $L_2$ norm) on $W$, we assume $w < 1$. With these two assumptions, the conclusion of theorem A.5 is rewritten as:

$$\text{mix}_{y_G}(b, a) \le \sum_{l=0}^{m_\Psi - 1} \left( \left( \boldsymbol{B}^{m_\Psi - l} \right)^\top diag \left( \boldsymbol{1}^\top \boldsymbol{B}^l \right) \boldsymbol{B}^{m_\Psi - l} \right)_{ab}. \tag{40}$$

With this new conclusion, we now turn to the proof of Theorem A.2:

*Proof.* Firstly, $diag \left( \boldsymbol{1}^\top \boldsymbol{B}^l \right)_i = (\alpha s^K)^l (((\hat{\boldsymbol{A}})^{K/2})^l \boldsymbol{1})_i \le \gamma (\alpha s^K)^l$ by using $(((\hat{\boldsymbol{A}})^{K/2})^l \boldsymbol{1})_i \le \gamma$ (Di Giovanni et al., 2023). Then, we find

$$\sum_{l=0}^{m_\Psi - 1} \left( \left( \boldsymbol{B}^{m_\Psi - l} \right)^\top diag \left( \boldsymbol{1}^\top \boldsymbol{B}^l \right) \boldsymbol{B}^{m_\Psi - l} \right)_{ab} \le \gamma \left( \sum_{l=0}^{m_\Psi - 1} \boldsymbol{B}^{2(m_\Psi - l)} \cdot (\alpha s^K)^l \right)_{ab}$$

$$< \gamma \left( \sum_{l=0}^{m_\Psi - 1} (\alpha(\hat{\boldsymbol{A}})^{K/2} s^K)^{2(m_\Psi - l)} \cdot (\alpha s^K)^l \right)_{ab}$$

$$< \gamma (\alpha s^K)^{2m_\Psi} \left( \sum_{l=0}^{m_\Psi - 1} \hat{\boldsymbol{A}}^{K(m_\Psi - l)} \right)_{ab}$$

$$= \gamma (\alpha s^K)^{2m_\Psi} \left( \sum_{l=1}^{m_\Psi} \hat{\boldsymbol{A}}^{Kl} \right)_{ab}. \tag{41}$$

The following proof depends on *commute time* $\tau(a, b)$ (Lovász, 1993), whose the definition is as follows using the spectral representation of the graph Laplacian (Di Giovanni et al., 2023):

$$\tau(a,b) = 2|E| \sum_{n=0}^{N-1} \frac{1}{\lambda_n} \left( \frac{u_n(a)}{\sqrt{d_a}} - \frac{u_n(b)}{\sqrt{d_b}} \right)^2.$$  (42)

Then, we have:

$$\left( \sum_{l=1}^{m_\Psi} \hat{\boldsymbol{A}}^{Kl} \right)_{ab} \leq \sum_{l=0}^{Km_\Psi} \left( \hat{\boldsymbol{A}}^l \right)_{ab}$$

$$= \sum_{l=0}^{Km_\Psi} \sum_{n \geq 0} (1 - \lambda_n)^l u_n(a) u_n(b)$$

$$= (Km_\Psi + 1) \frac{\sqrt{d_a d_b}}{2|E|} + \sum_{n>0} \frac{1 - (1-\lambda)^{Km_\Psi+1}}{\lambda_n} u_n(a) u_n(b)$$  (43a)

$$= (Km_\Psi + 1) \frac{\sqrt{d_a d_b}}{2|E|} + \sum_{n>0} \frac{1}{\lambda_n} u_n(a) u_n(b) - \sum_{n>0} \frac{(1-\lambda)^{Km_\Psi+1}}{\lambda_n} u_n(a) u_n(b).$$

In Eq. (43a), we use $u_0(a) = \sqrt{\frac{d_a}{2|E|}}$. Then, from the definition of commute time, we can get:

$$\sum_{n=1}^{N-1} \frac{1}{\lambda_n} u_n(a) u_n(b) = \frac{-\tau(a,b)}{4|E|} \sqrt{d_a d_b} + \frac{1}{2} \sum_{n>0} \frac{1}{\lambda_n} \left( u_n^2(a) \sqrt{\frac{d_b}{d_a}} + u_n^2(b) \sqrt{\frac{d_a}{d_b}} \right)$$

$$\leq \frac{-\tau(a,b)}{4|E|} \sqrt{d_a d_b} + \frac{1}{2\lambda_1} \left( \sqrt{\frac{d_a}{d_b}} + \sqrt{\frac{d_b}{d_a}} - \frac{\sqrt{d_a d_b}}{|E|} \right),$$  (44)

where in the last inequation, we utilize the fact that $\sum_{n>0} u_n^2(a) = 1 - u_0^2(a)$ because $\{u_n\}$ is a set of orthonormal basis. Besides, we use $\lambda_n > \lambda_1, \forall n > 1$. Next, we derive

$$-\sum_{n>0} \frac{(1-\lambda)^{Km_\Psi+1}}{\lambda_n} u_n(a) u_n(b) \leq \sum_{n>0} \frac{|1-\lambda^*|^{Km_\Psi+1}}{\lambda_n} |u_n(a) u_n(b)||$$

$$\leq \frac{|1-\lambda^*|^{Km_\Psi+1}}{2\lambda_1} \sum_{n>0} (u_n^2(a) + u_n^2(b))$$  (45)

$$\leq \frac{|1-\lambda^*|^{Km_\Psi+1}}{2\lambda_1} \left( 2 - \frac{d_a + d_b}{2|E|} \right),$$

where $|1-\lambda^*| = \max_{0 < n \leq N-1} |1 - \lambda_n| < 1$. Insert derivations (44) and (45) into (43), then gather all above derivations:

$$\text{mix}_{y_G}(b, a) \leq \gamma(\alpha s^K)^{2m_\Psi} \left\{ (Km_\Psi + 1) \frac{\sqrt{d_a d_b}}{2|E|} - \frac{\tau(a,b)}{4|E|} \sqrt{d_a d_b} \right.$$

$$\left. + \frac{1}{2\lambda_1} \left( \sqrt{\frac{d_a}{d_b}} + \sqrt{\frac{d_b}{d_a}} - \frac{\sqrt{d_a d_b}}{|E|} \right) + \frac{|1-\lambda^*|^{Km_\Psi+1}}{2\lambda_1} \left( 2 - \frac{d_a + d_b}{2|E|} \right) \right\}$$

$$\leq \gamma(\alpha s^K)^{2m_\Psi} \sqrt{d_a d_b} \left( \frac{Km_\Psi}{2|E|} - \frac{\tau(a,b)}{4|E|} \right) + \frac{\gamma(\alpha s^K)^{2m_\Psi}}{2\lambda_1} \left( \sqrt{\frac{d_a}{d_b}} + \sqrt{\frac{d_b}{d_a}} \right) + \frac{\gamma(\alpha s^K)^{2m_\Psi}}{\lambda_1} |1-\lambda^*|^{Km_\Psi+1}.$$  (46)

In last inequation, we discard $\frac{\sqrt{d_a d_b}}{2|E|} \left[ 1 - \frac{1}{\lambda_1} \left( 1 + \frac{|1-\lambda^*|^{Km_\Psi+1}}{2} \left( \sqrt{\frac{d_a}{d_b}} + \sqrt{\frac{d_b}{d_a}} \right) \right) \right] < 0$ because $\lambda_1 < 1$. Then,

$$\frac{\text{mix}_{y_G}(b, a)}{\gamma(\alpha s^K)^{2m_\Psi} \sqrt{d_a d_b}} \leq \frac{Km_\Psi}{2|E|} - \frac{\tau(a,b)}{4|E|} + \frac{1}{2\lambda_1 \sqrt{d_a d_b}} \left( \sqrt{\frac{d_a}{d_b}} + \sqrt{\frac{d_b}{d_a}} + 2|1-\lambda^*|^{Km_\Psi+1} \right).$$  (47)

From (47), we can finally give the lower bound of $m_\Psi$ as:

$$m_\Psi \geq \frac{2|E|}{K} \left\{ \frac{\tau(a,b)}{4|E|} + \frac{\text{mix}_{y_G}(b,a)}{\gamma(\alpha s^K)^{2m_\Psi}\sqrt{d_a d_b}} - \frac{1}{2\lambda_1\sqrt{d_a d_b}} \left( \sqrt{\frac{d_a}{d_b}} + \sqrt{\frac{d_b}{d_a}} + 2|1-\lambda^*|^{Km_\Psi+1} \right) \right\}$$

$$> \frac{2|E|}{K} \left\{ \frac{\tau(a,b)}{4|E|} + \frac{1}{\sqrt{d_a d_b}} \left[ \frac{\text{mix}_{y_G}(b,a)}{\gamma(\alpha s^K)^{2m_\Psi}} - \frac{1}{2\lambda_1} \left( 2\gamma + 2|1-\lambda^*|^{Km_\Psi+1} \right) \right] \right\}$$

$$= \frac{2|E|}{K} \left\{ \frac{\tau(a,b)}{4|E|} + \frac{1}{\sqrt{d_a d_b}} \left[ \frac{\text{mix}_{y_G}(b,a)}{\gamma(\alpha^2 s^{2K})^{m_\Psi}} - \frac{1}{\lambda_1} \left( \gamma + |1-\lambda^*|^{Km_\Psi+1} \right) \right] \right\}$$

$$= \frac{\tau(a,b)}{2K} + \frac{2|E|}{K\sqrt{d_a d_b}} \left[ \frac{\text{mix}_{y_G}(b,a)}{\gamma(\alpha^2 s^{2K})^{m_\Psi}} - \frac{1}{\lambda_1} \left( \gamma + |1-\lambda^*|^{Km_\Psi+1} \right) \right]$$

$$\tag{48}$$

$\square$

### A.3 PROOF OF THEOREM 4.2 (SHORT-RANGE AND LONG-RANGE RECEPTIVE FIELDS)

*Proof.* From theorem A.2, we denote $L_{m_\Psi} = \frac{\tau(a,b)}{2K} + \frac{2|E|}{K\sqrt{d_a d_b}} \left[ \frac{\text{mix}_{y_G}(b,a)}{\gamma(\alpha^2 s^{2K})^{m_\Psi}} - \frac{1}{\lambda_1} \left( \gamma + |1-\lambda^*|^{Km_\Psi+1} \right) \right]$. For K-order message passing $\sigma(\sum_{j=0}^K \tau_j A^j HW)$, $\tau_j \in [0,1]$, we assume that $(\tau_P A^P)_{ba}$ is the maximum among $\{(\tau_0 A^0)_{ba}, \ldots, (\tau_K A^K)_{ba}\}$. According to theorem A.5, we can get the similar conclusion, replacing $B$ with $C = (K+1)\tau_P A^P$. Then, we have the following proof:

*Proof.* Again, $\text{diag}\left(\mathbf{1}^\top C^l\right)_i = ((K+1)\tau_P)^l (A^{Pl}\mathbf{1})_i \leq \gamma((K+1)\tau_P)^l$. Then, we have

$$\sum_{l=0}^{m_A-1} \left( \left(C^{m_A-l}\right)^\top \text{diag}\left(\mathbf{1}^\top C^l\right) C^{m_A-l} \right)_{ab} \leq \gamma \left( \sum_{l=0}^{m_A-1} C^{2(m_A-l)} \cdot ((K+1)\tau_P)^l \right)_{ab}$$

$$< \gamma \left( \sum_{l=0}^{m_A-1} ((K+1)\tau_P A^P)^{2(m_A-l)} \cdot ((K+1)\tau_P)^l \right)_{ab}$$

$$< \gamma((K+1)\tau_P)^{2m_A} \left( \sum_{l=0}^{m_A-1} \hat{A}^{2P(m_A-l)} \right)_{ab}$$

$$= \gamma((K+1)\tau_P)^{2m_A} \left( \sum_{l=1}^{m_A} \hat{A}^{2Pl} \right)_{ab}$$

$$< \gamma(\sqrt{(K+1)\tau_P})^{4m_A} \left( \sum_{l=1}^{2m_A} \hat{A}^{Pl} \right)_{ab}.$$

$$\tag{49}$$

$\square$

Following the rest proof of $L_{m_\Psi}$, replace $\{\alpha s^K, m_\Psi, K\}$ with $\{\sqrt{(K+1)\tau_P}, 2m_A, P\}$, and get the expression of $L_{m_A}$:

$$L_{m_A} = \frac{\tau(a,b)}{2P} + \frac{2|E|}{P\sqrt{d_a d_b}} \left[ \frac{\text{mix}_{y_G}(b,a)}{\gamma((K+1)^2\tau_P^2)^{m_A}} - \frac{1}{\lambda_1} \left( \gamma + |1-\lambda^*|^{2Pm_A+1} \right) \right]. \tag{50}$$

Therefore, we have

$$L_{m_\Psi} \approx \frac{P}{K} L_{m_A} + \frac{2|E|}{K\sqrt{d_a d_b}} \left[ \frac{\text{mix}_{y_G}(b,a)}{\gamma} \left( \frac{1}{(\alpha^2 s^{2K})^{m_\Psi}} - \frac{1}{((K+1)^2\tau_P^2)^{m_A}} \right) \right], \tag{51}$$

where we ignore $|1-\lambda^*|^{Km_\Psi+1}$ and $|1-\lambda^*|^{2Pm_A+1}$. Since $|1-\lambda^*| < 1$ as shown in theorem A.2, therefore $|1-\lambda^*|^{Km_\Psi+1} - |1-\lambda^*|^{2Pm_A+1}$ will be very small, especially when $m_\Psi$ and $m_A$ are large. From Eq. (51), when $s \to \infty$, the relation becomes:

$$L_{m_\Psi} \approx \frac{P}{K} L_{m_A} - \frac{2|E|}{K(K+1)^{2m_A}\tau_P^{2m_A}\sqrt{d_a d_b}} \frac{\text{mix}_{y_G}(b,a)}{\gamma}. \tag{52}$$

Or, when $s \to 0$, the relation becomes:

$$L_{m_\Psi} \approx \frac{P}{K} L_{m_A} + \frac{2|E|}{K\sqrt{d_a d_b}} \frac{\text{mix}_{y_G}(b, a)}{\gamma} \cdot \frac{1}{(\alpha^2 s^{2K})^{m_\Psi}}. \tag{53}$$

$\square$

## B RELATIONSHIP BETWEEN GRAPH CONVOLUTION AND TRANSFORMER

To recap, the vanilla Transformer (Vaswani et al., 2017) is given as:

$$\boldsymbol{H}^{(l+1)} = \text{softmax}\left(\frac{\boldsymbol{H}^{(l)}\boldsymbol{W}_q(\boldsymbol{H}^{(l)}\boldsymbol{W}_k)^\top}{\sqrt{d}}\right)\boldsymbol{H}^{(l)}\boldsymbol{W}_v, \tag{54}$$

where $\boldsymbol{H}^{(l)}, \boldsymbol{H}^{(l+1)} \in \mathbb{R}^{N \times d}$ are node embeddings from $l$th and $(l+1)$th layers, and $\{\boldsymbol{W}_q, \boldsymbol{W}_k, \boldsymbol{W}_v\} \in \mathbb{R}^{d \times d}$ are the query, key, and value learnable matrices. This self-attention mechanism can be written as a kernel summation in the discrete case (Mialon et al., 2021; Tsai et al., 2019; Guibas et al., 2021). Specifically for node $s$, $\boldsymbol{h}_s^{(l+1)} = \sum_t \boldsymbol{\kappa}(s,t)\boldsymbol{h}_t^{(l)}$, where $\boldsymbol{\kappa}(s,t) = \text{softmax}(\frac{\boldsymbol{H}^{(l)}\boldsymbol{W}_q(\boldsymbol{H}^{(l)}\boldsymbol{W}_k)^\top}{\sqrt{d}})_{(s,t)} \cdot \boldsymbol{W}_v$. Therefore, $\boldsymbol{\kappa} : \{1, \dots, N\} \times \{1, \dots, N\} \to \mathbb{R}^{d \times d}$ is treated as an asymmetric matrix-valued kernel. Note that the usage of asymmetric kernel is also commonly used in various machine learning tasks (Kulis et al., 2011). Further assume $\boldsymbol{\kappa}(s,t) = \boldsymbol{\kappa}(s-t)$, which indicates a shift-invariant GT since the attention depends on the difference between two nodes rather than their positions. Then, Eq. (54) becomes a convolution $\boldsymbol{h}_s^{(l+1)} = \sum_t \boldsymbol{\kappa}(s-t)\boldsymbol{h}_t^{(l)}$, which can be expressed with the convolution theorem as:

$$\boldsymbol{h}_s^{(l+1)} = \mathcal{F}^{-1}(\mathcal{F}(\boldsymbol{\kappa}) \cdot \mathcal{F}(\boldsymbol{H}^{(l)}))(s) \in \mathbb{R}^{1 \times d}. \tag{55}$$

Eq. (55) is also known as Fourier integral operator (Hörmander, 1971). In Eq. (55), for each frequency mode $n \in N$ (i.e., $\mathcal{F}(\boldsymbol{H}^{(l)})(n, \cdot)$), $\mathcal{F}(\boldsymbol{\kappa})(n) \in \mathbb{R}^{d \times d}$, because $\boldsymbol{\kappa} : \{1, \dots, N\} \times \{1, \dots, N\} \to \mathbb{R}^{d \times d}$. Hence, for all modes, $\mathcal{F}(\boldsymbol{\kappa})$ can be fully parameterized by a neural network $R_\theta \in \mathbb{R}^{N \times d \times d}$ (Li et al., 2021):

$$\boldsymbol{h}_s^{(l+1)} = \mathcal{F}^{-1}(R_\theta \cdot \mathcal{F}(\boldsymbol{H}^{(l)}))(s) \in \mathbb{R}^{1 \times d}. \tag{56}$$

## C DETAILS OF ENCODING EIGENVALUES

In this paper, we adopt Eigenvalue Encoding (EE) Module (Bo et al., 2023) to encode eigenvalues. EE functions as a set-to-set spectral filter, enabling interactions between eigenvalues. In EE, both magnitudes and relative differences of all eigenvalues are leveraged. Specifically, the authors use an eigenvalue encoding function to transform each $\lambda$ from scalar $\mathbb{R}^1$ to a vector $\mathbb{R}^d$:

$$\rho(\lambda, 2i) = \sin\left(\epsilon\lambda/10000^{2i/d}\right), \quad \rho(\lambda, 2i+1) = \cos\left(\epsilon\lambda/10000^{2i/d}\right), \tag{57}$$

where $i$ is the dimension of the representations and $\epsilon$ is a hyper parameter. By encoding in this way, relative frequency shifts between eigenvalues are captured. Then, the raw representations of eigenvalues are the concatenation between eigenvalues and corresponding representation vectors:

$$\boldsymbol{Z}_\lambda = [\lambda_1||\rho(\lambda_1), \dots, \lambda_N||\rho(\lambda_N)]^\top \in \mathbb{R}^{N \times (d+1)}. \tag{58}$$

To capture the dependencies between eigenvalues, a standard Transformer is used followed by skip-connection and feed forward network (FFN):

$$\hat{\boldsymbol{Z}}_\lambda = \text{Transformer}(\text{LN}(\boldsymbol{Z}_\lambda)) + \boldsymbol{Z}_\lambda \in \mathbb{R}^{N \times (d+1)}, \quad \boldsymbol{Z} = \text{FFN}(\text{LN}(\hat{\boldsymbol{Z}}_\lambda)) + \hat{\boldsymbol{Z}}_\lambda \in \mathbb{R}^{N \times (d+1)}, \tag{59}$$

where LN is the layer normalization. Then, $\boldsymbol{Z}$ is the embedding matrix for eigenvalues, which is injected into the learning of combination coefficients $\tilde{\boldsymbol{a}}$ and $\tilde{\boldsymbol{b}}$, and scales $\tilde{\boldsymbol{s}}$.

# D EXPERIMENTAL DETAILS

## D.1 IMPLEMENTATION DETAILS

For the short-range task, we present the results of all baselines from NAGphormer (Chen et al., 2022b) and Exphormer (Shirzad et al., 2023), excluding three base models (Transformer, SAN, and GraphGPS). This omission is due to the necessity of knowing the parameters of these base models for parameter freezing, while such information is unavailable. Consequently, we provide the results of these three models based on our experiments. For the long-range task, we showcase the outcomes of GCN, GINE, GatedGCN, Transformer, SAN+LapPE, SAN+RWSE from (Dwivedi et al., 2022), alongside the results of the remaining baselines sourced from their original papers.

Considering that our WaveGC replaces the Transformer in each base model, we focus on tuning the parameters newly introduced by WaveGC while keeping the others unchanged. Specifically, we explore the number of truncated terms $\rho$ from 1 to 10 and adjust the number of scales $J$ from 1 to 5. Additionally, for the pre-defined vector $\overline{s}$ controlling the amplitudes of scales, we test each element in $\overline{s}$ from 0.1 to 10. The usage of the tight frames constraint is also a parameter subject to tuning, contingent on the given dataset. Typically, models iterate through several layers to produce a single result, thus the parameters of WaveGC may or may not be shared between different layers. Finally, due to the large scale of short-range datasets, we implement two strategies to prevent out-of-memory issues. Firstly, only the first 30% of eigenvalues and their corresponding eigenvectors are retained for training in each dataset. Secondly, for the learned scaling function basis $\Phi$ and multiple wavelet bases $\Psi_{s_j}$, we set a threshold $\aleph$ and filter out entries in $\Phi$ and $\Psi_{s_j}$ whose absolute value is lower than $\aleph$.

For fair comparisons, we randomly run 4 times on long-range datasets (Dwivedi et al., 2022), and 10 times on short-range datasets (Chen et al., 2022b), and report the average results with their standard deviation for all methods. For the sake of reproducibility, we also report the related parameters in Appendix D.9.

## D.2 DATASETS DESCRIPTION

Table 8: The statistics of the short-range datasets.

| Dataset | # Graphs | # Nodes | # Edges | # Features | # Classes |
|---------|----------|---------|---------|------------|-----------|
| CS | 1 | 18,333 | 163,788 | 6,805 | 15 |
| Photo | 1 | 7,650 | 238,163 | 745 | 8 |
| Computer | 1 | 13,752 | 491,722 | 767 | 10 |
| CoraFull | 1 | 19,793 | 126,842 | 8,710 | 70 |
| ogbn-arxiv | 1 | 169,343 | 1,116,243 | 128 | 40 |

For short-range datasets, we choose five commonly used `CS`, `Photo`, `Computer`, `CoraFull` (Fey & Lenssen, 2019) and `ogbn-arxiv` (Hu et al., 2020). `CS` is a network based on co-authorship, with nodes representing authors and edges symbolizing collaboration between them. In the `Photo` and `Computer` networks, nodes stand for items, and edges suggest that the connected items are often purchased together, forming co-purchase networks. `CoraFull` is a network focused on citations, where nodes are papers and edges indicate citation connections between them. `ogbn-arxiv` is a citation network among all Computer Science (CS) Arxiv papers, where each node corresponds to an Arxiv paper, and the edges indicate the citations between papers. The details of these five datasets are summarized in Table 8.

Table 9: The statistics of the long-range datasets.

| Dataset | # Graphs | Avg. # nodes | Avg. # edges | Prediction level | Task | Metric |
|---------|----------|--------------|--------------|------------------|------|--------|
| PascalVOC-SP | 11,355 | 479.4 | 2,710.5 | inductive node | 21-class classif. | F1 score |
| PCQM-Contact | 529,434 | 30.1 | 61.0 | inductive link | link ranking | MRR |
| COCO-SP | 123,286 | 476.9 | 2,693.7 | inductive node | 81-class classif. | F1 score |
| Peptides-func | 15,535 | 150.9 | 307.3 | graph | 10-task classif. | Avg. Precision |
| Peptides-struct | 15,535 | 150.9 | 307.3 | graph | 11-task regression | Mean Abs. Error |

For long-range tasks, we choose five long-range datasets (Dwivedi et al., 2022), including `PascalVOC-SP` (VOC), `PCQM-Contact` (PCQM), `COCO-SP`(COCO), `Peptides-func` (Pf) and `Peptides-struct` (Ps). These five datasets are usually used to test the performance of different transformer architectures. `VOC` and `COCO` datasets are created through SLIC superpixelization of the Pascal VOC and MS COCO image collections. They are both utilized for node classification, where each super-pixel node is categorized into a specific object class. `PCQM` is developed from PCQM4Mv2 (Hu et al., 2021) and its related 3D molecular structures, focusing on binary link prediction. This involves identifying node pairs that are in 3D contact but distant in the 2D graph. Both `Pf` and `Ps` datasets consist of atomic graphs of peptides sourced from SATPdb. In the `Peptides-func` dataset, the task involves multi-label graph classification into 10 distinct peptide functional classes. Conversely, the `Peptides-struct` dataset is centered on graph regression to predict 11 different 3D structural properties of peptides. The details of these five datasets are summarized in Table 9.

### D.3 MORE ANALYSES FOR SECTION 6.2

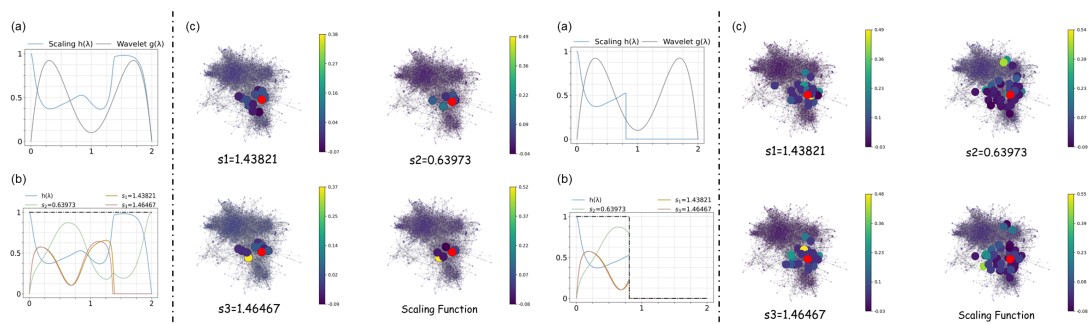

Figure 3: Illustration of the spectral and spatial signals of the learned function basis and multiple wavelet bases with full spectrum.

Figure 4: Illustration of the spectral and spatial signals of the learned function basis and multiple wavelet bases with partial spectrum.

In this section, we provide more analyses on the visualization performances for different bases. Upon examination of three kinds of wavelets (i.e., {SGWT, DEFT, GWNN}), those from SGWT meet admissibility criteria with multiple resolutions, but these cubic splines are not adaptive. DEFT outputs several bases with unpredictable shapes, so it is hard to strictly restrain these outputs as wavelets. GWNN adopts one exponential wavelet base, omitting information from different ranges as well as not meeting criteria. The following three polynomial bases (i.e., ChebNet*, ChebNetII and JacobiConv) comprehensively entangles signals from different frequency intervals, where crucial band-pass signals for long-range tasks are overwhelmed. Consequentially, these bases blend local and distant information in spatial space, hampering the decision on the best range. For our WaveGC, Fig. 2 intuitively demonstrates that the unit wavelet got by our decoupling of Chebyshev polynomials strictly meets the admissibility criteria, as Eq. equation 1, while the corresponding base scaling function supplements the direct current signals at $\lambda = 0$. After integration of learnable scales, the final wavelets also meet criteria and adapt to the demand on multiresolution. The plot of $G(\lambda) = h(\lambda)^2 + \sum_{j=1}^{3} g(s_j \lambda)^2$ as a black dashed line (located at 1) confirms the construction of tight frames via normalization technique. Fig.2 also depicts the signal distribution over the topology centered on the target node (the red-filled circle). This figure also demonstrates that as the scale $s_j$ increases, the receptive field of the central node expands. Once again, this visualization intuitively confirms the capability of WaveGC to aggregate both short- and long-range information simultaneously but distinguishingly.

To give one more example, we provide additional visualization results on the CoraFull dataset. These results are presented in Fig. 3, where the learned scaling functions $h(\lambda)$ and $g(\lambda)$ meet the specified requirements. The four subfigures in Fig. 3(c) illustrate that as the scale $s_j$ increases, the receptive field of the center node expands. This highlights WaveGC's capability to capture both short- and long-range information by adjusting different values of $s_j$. However, one of our strategies for CoraFull involves considering only 30% of eigenvalues as input. Consequently, the full spectrum is truncated,

leaving only the remaining 30% parts, as depicted in Fig. 4. As shown in Fig. 4(b), all $g(s_j\lambda)$ functions behave like band-pass filters with large $s_j$ values due to this truncation. Consequently, all three learned wavelets enable the center node to receive distant information, as demonstrated in Fig. 4(c).

## D.4 MORE ABLATION STUDY

Table 10: Results of the ablation study. **Bold**: Best.

| Variants | Pf | VOC | Computer | CoraFull |
| --- | --- | --- | --- | --- |
| | AP ↑ | F1 score ↑ | Accuracy ↑ | Accuracy ↑ |
| w/o $h(\lambda)$ | 56.56 | 15.69 | 89.37 | 63.36 |
| w/o $g(s_j\lambda)$ | 61.89 | 25.51 | 89.67 | 63.78 |
| w/o tight frame | 64.71 | 27.79 | **89.77** | **64.74** |
| **Ours** | **65.18** | **29.42** | 89.69 | 63.97 |

Table 11: Different combinations between WaveGC GCN and Transformer. **Bold**: Best.

| Combinations | Pf | Photo |
| --- | --- | --- |
| | AP ↑ | Accuracy ↑ |
| MPGNN + Transformer | 94.47 | 65.35 |
| WaveGC + Transformer | 95.04 | 67.57 |
| WaveGC + MPGNN | **95.37** | **70.10** |

In this section, we conduct an ablation study of our WaveGC to assess the effectiveness of each component, and the corresponding results are presented in Table 10. The table compares Transformer+WaveGC with several variants. These variants involve removing the scaling function basis (denoted as 'w/o $h(\lambda)$'), excluding multiple wavelet bases (denoted as w/o '$g(s_j\lambda)$'), or eliminating the constraint on tight frames (denoted as 'w/o tight frame'). The evaluation is conducted on two long-range datasets (`Pf` and `VOC`) and two short-range datasets (`Computer` and `CoraFull`). (1) Both the scaling function basis $h(\lambda)$ and wavelet bases $g(s_j\lambda)$ are essential components of our WaveGC. In particular, neglecting $h(\lambda)$ results in a significant drop in performance, emphasizing the crucial role of low-frequency information. (2) The tight frame constraint proves beneficial for Pf and VOC datasets but is less effective for Computer and CoraFull. This suggests a trade-off, as the tight frame constraint limits the flexibility of the learned filters.

This work involves three frameworks, including MPGNN (graph convolution), Transformer and WaveGC, and exploring the benefits of combination between these frameworks is also an interesting topic. The related results are given in Table 11. Upon comparing the first two combinations in the table, 'Transformer' primarily focuses on capturing global information, while 'MPGNN' or 'WaveGC' are expected to focus on local information. Given that MPGNN is proficient in depicting local structure, the improvement from WaveGC is somewhat limited. However, in the third combination 'MPGNN+WaveGC', WaveGC is designed to capture both local and global information. The noticeable improvement compared to 'MPGNN+Transformer' can be attributed to the flexibility and multi-resolution capabilities of WaveGC. In summary, both MPGNN and WaveGC are effective at capturing local structure, while WaveGC excels in encoding long-range information. For practical applications, it is advisable to select the specific encoder based on the given graph.

Table 12: More ablations for differences between WaveGC and ChebNet.

| | Free $\tilde{\alpha}$ | Free $\tilde{\beta}$ | Fix s=1 | Free $\tilde{s}$ | Original |
| --- | --- | --- | --- | --- | --- |
| Computer | 90.35±0.07 | 90.30±0.12 | 90.55±0.02 | 90.32±0.01 | **91.00±0.48** |
| Ps | 25.08±0.01 | 25.09±0.12 | 25.28±0.00 | 25.15±0.25 | **24.95±0.07** |

Obviously, both WaveGC and ChebNet attempt weighted combination of Chebyshev polynomials in different ways. On one hand, ChebNet learns term coefficients independently, while WaveGC map eigenvectors into coefficients $\tilde{\alpha}$ and $\tilde{\beta}$. On the other hand, WaveGC further involve multiple and learnable scales $\tilde{s}$. Finally, we test importance of these differences on the Computer and Ps. The results are summarized in Table 12, showcasing different variants such as free learning coefficients (i.e., $\tilde{\alpha}$, $\tilde{\beta}$), adopting single scale s=1, and free learning $\tilde{s}$ to avoid joint parameterization. Each of these modifications resulted in degraded performance compared to the original model, demonstrating the improvements our new model offers over ChebNet.

## D.5 Time and space complexity analysis

In this section, we report the running time and GPU memory consumption of the three base models and their corresponding WaveGC versions. The results for Photo and PCQM are presented in Table 13. According to the table, the resource consumption of WaveGC is nearly the same as that of the base models. Specifically, SAN+WaveGC runs faster and uses less GPU memory than SAN on PCQM because SAN also includes edge features when calculating attention. Due to memory limitations, we ran SAN in sparse mode on Photo, where the attention range is limited to one-hop rather than the full graph. This accounts for its efficiency on Photo. Moreover, WaveGC consumes less memory on Photo compared to GPS and the vanilla Transformer. This advantage stems from using only a subset of the eigenvalues and eigenvectors.

Table 13: Comparison on running time and GPU memory consumption.

| Datasets | Consumption | GPS | WaveGC+GPS | SAN | WaveGC+SAN | Transformer | WaveGC+Transformer |
|---|---|---|---|---|---|---|---|
| Photo | times / epoch (s) | 0.33 | 0.72 | 0.28 | 1.66 | 0.25 | 0.65 |
| | GPU memory (MB) | 5837 | 4829 | 5559 | 11691 | 5733 | 4783 |
| PCQM | times / epoch (s) | 326 | 473 | 867 | 484 | 295 | 402 |
| | GPU memory (MB) | 1647 | 4229 | 17499 | 16137 | 1423 | 2035 |

For a deeper investigation, we measure the growth trend of space and computational complexity with 1) increasing the graph size and 2) increasing the layer depth. For the first scenario, we used ogbn-arxiv as the reference graph and constructed a series of subgraphs of varying sizes by retaining different ratios of nodes. For the second scenario, we used a subgraph containing 50% of the nodes as the base graph and varied the model layers from 1 to 10. The results are illustrated in Fig. 5, leading to the following conclusions: (1) As the graph size and layer depth increase, both the time and space complexities of WaveGC scale linearly. (2) While the time difference between WaveGC and Performer remains nearly constant as the graph size increases, increasing the layer depth widens their time gap. (3) In both scenarios, WaveGC uses less memory than Performer, and this memory advantage becomes more pronounced as the graph size and number of layers increase.

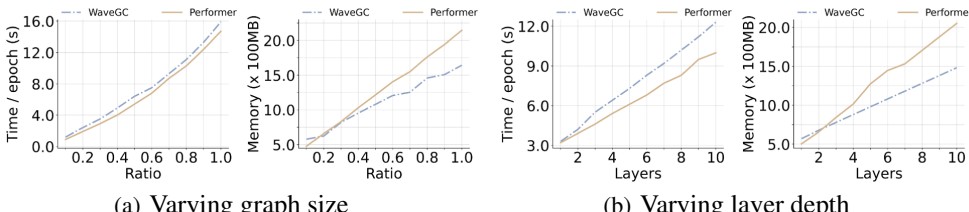

(a) Varying graph size      (b) Varying layer depth

Figure 5: The time and memory grows with increasing graph size and layer depth.

## D.6 Experimental results on heterophily datasets

Besides the short-rang and long-range tasks, heterophily benchmark datasets are also important scenarios for testing graph spectral methods. Here, we choose three heterophily datasets, including Actor (Pei et al., 2020),

Table 14: The statistics of the heterophily datasets.

| Dataset | # Graphs | # Nodes | # Edges | # Features | # Classes |
|---|---|---|---|---|---|
| Actor | 1 | 7,600 | 33,544 | 932 | 5 |
| Minesweeper | 1 | 10,000 | 39,402 | 7 | 2 |
| Tolokers | 1 | 11,758 | 519,000 | 10 | 2 |

Minesweeper and Tolokers (Platonov et al., 2023). Table 14 shows their basic statistics. We firstly compare WaveGC with corresponding base models (Table 15), and then compare with other graph Transformers (Table 16), including {Graphormer, Nodeformer, Specformer, SGFormer, NAG-phormer and Exphormer}. Again, our WaveGC outperforms both base models and other powerful graph Transformers. Especially, for Minesweeper and Tolokers, WaveGC also defeats other baselines reported in the original paper (Platonov et al., 2023).

The superiority on heterophily owes to the "scale effect" of $s$ mentioned in section 3.1. Thus, if WaveGC can finally learn $s < 1$, $g(s\lambda) > 0$ when $\lambda = 2$ ("stretch" effect), so WaveGC will maintain high-frequency signals and achieve better performance on heterophily graphs. To illustrate, we plotted the graph spectrum on Actor in Fig. 6, and showed that indeed $s < 1$ for two learned scales with corresponding high-pass filters. Please note $g(s\lambda)$ still belongs to the class of graph wavelets in this case, since it always meets the wavelet admissibility criteria in Eq. 1.

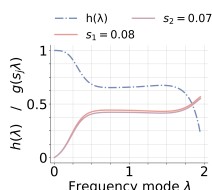

Figure 6: Spectrum of Actor.

Table 15: Quantified results (i.e. $\%\pm\sigma$) on heterophily datasets compared to base models.

| Model | Transformer | w/ WaveGC | SAN | w/ WaveGC | GraphGPS | w/ WaveGC |
|---|---|---|---|---|---|---|
| Actor (Accuracy ↑) | 37.63±0.55 | **38.61±0.74** | 31.18±1.08 | **33.63±0.82** | 36.52±0.56 | **37.40±1.04** |
| Minesweeper (ROC AUC ↑) | 50.75±1.14 | **93.19±1.56** | 92.07±0.35 | **93.98±0.60** | 94.03±0.42 | **94.81±0.42** |
| Tolokers (ROC AUC ↑) | 74.04±0.53 | **82.81±1.12** | 83.37±0.55 | 82.73±0.98 | 84.63±0.88 | **85.38±0.52** |

Table 16: Qualified results (i.e. $\%\pm\sigma$) on heterophily tasks compared to baselines. **Bold**: Best, Underline: Runner-up, '*' Taken from original papers.

| Model | Graphormer | Nodeformer | Spcformer | SGFormer | NAGphormer | Exphormer | **WaveGC** |
|---|---|---|---|---|---|---|---|
| Actor (Accuracy ↑) | 36.41±0.49 | 34.62±0.82 | **41.93±1.04*** | 37.90±1.10 | 35.39±0.80 | 36.45±1.21 | 38.61±0.74 |
| Minesweeper (ROC AUC ↑) | 90.89±0.53 | 86.71±0.88 | 89.93±0.41 | 94.31±0.41 | 88.06±0.43 | 90.57±0.64 | **94.81±0.42** |
| Tolokers (ROC AUC ↑) | 82.75±0.88 | 78.10±1.03 | 80.42±0.55 | 84.57±0.70 | 82.02±0.98 | 84.68±0.77 | **85.38±0.52** |

### D.7 HYPER-PARAMETER SENSITIVITY ANALYSIS

In WaveGC, two key hyper-parameters, namely $\rho$ and $J$, play important roles. The parameter $\rho$ governs the number of truncated terms for both $T_i^o$ and $T_i^e$, while $J$ determines the number of scales $s_j$ in Eq. equation 7. In this section, we explore the sensitivity of $\rho$ and $J$ on the Peptides-struct (Ps) and Computer datasets. The results are visually presented in Fig.7, where the color depth of each point reflects the corresponding performance (the lighter the color, the better the performance), and the best points are identified with a red star. Observing the results, we note that the optimal value for $\rho$ is 2 for Ps and 7 for Computer. This discrepancy can be attributed to the substantial difference in the graph sizes between the two datasets, with Computer exhibiting a significantly larger graph size (refer to Appendix D.2). Consequently, a more intricate filter design is necessary for the larger dataset. Concerning $J$, the optimal value is determined to be 3 for both Ps and Computer. A too small $J$ leads to inadequate coverage of ranges, while an excessively large $J$ results in redundant scales with overlapping ranges.

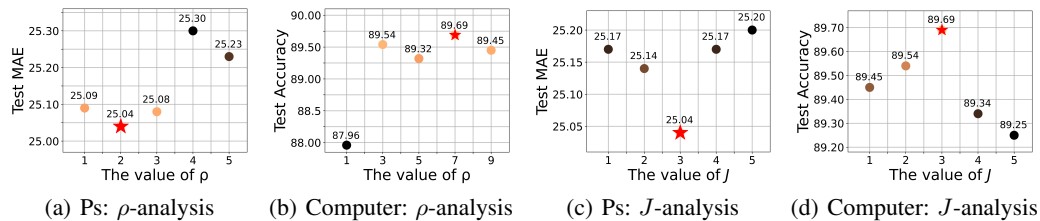

(a) Ps: $\rho$-analysis   (b) Computer: $\rho$-analysis   (c) Ps: $J$-analysis   (d) Computer: $J$-analysis

Figure 7: Analysis of the sensitivities of $\rho$ and $J$.

### D.8 DETAILED DESCRIPTIONS OF BASE MODELS

As introduced in experiments, we choose three base models, including Transformer (Vaswani et al., 2017), SAN (Kreuzer et al., 2021) and GraphGPS (Rampásek et al., 2022), and then replace the Transformer component with our WaveGC for each model, to roundly verify the effectiveness of WaveGC. Therefore, it is necessary to briefly introduce their mechanisms in this section. The illustrations of these three methods are given in Fig. 8.

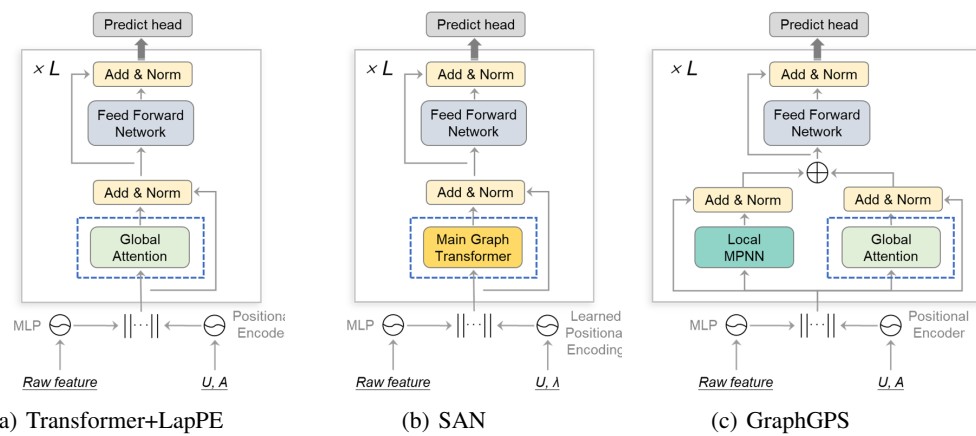

Figure 8: Illustrations of the three base models, including Transformer+LapPE, SAN and GraphGPS. The component surrounded by blue dash line is the part that will be replaced by WaveGC.

- **Transformer.** In Fig. 8(a), we show the case that the first several eigenvectors are regarded as positional encoding for each node, denoted as LapPE. During the training, the sign of these eigenvectors is frequently flipped to manipulate the model to avoid the influence from the original sign. Then, this positional encoding is concatenated with raw feature after projection, and input to a vanilla Transformer (Global Attention). The output of Transformer will be combined with its input, and pass through two-layers MLP (Feed Forward Network). After another skip-connection and normalization, we finish one layer of Transformer. In traditional settings, this process will loop several times.

- **SAN.** As shown in Fig. 8(b), SAN proposes some special designs for both positional encoding and Transformer. For the former, Learned positional encoding (LPE) architecture is given. Specifically, for some node $i$, the authors concatenate the eigenvalues and $i$ th entry in each normalized eigenvector, then they use linear layer to mix eigenvalues and eigenvectors, and then a Transformer encoder is utilized to mix different channels. Finally, a sum pooling layer is utilized to get the final LPE for each node. For the latter, similar to Transformer, the authors additionally consider the effect of edge feature when calculating attention values. To emphasize the importance of local structure, they assign different weights to neighbor nodes and distant nodes.

- **GraphGPS.** The aim of this work is to build a general, powerful, scalable grpah Transformer with linear complexity as shown in Fig. 8(c). Besides the global attention part, they explicitly involve a parallel MPNN to encode the given topological structure. Then, these two branches separately go through skip-connection and normalization, and then sum together followed by FFN, skip-connection and normalization. For generalization, the authors also provide different choices for positional encoding, local MPNN and Global Attention.

The Global Attention (or Main Graph Transformer) part is surrounded by blue dash line, which is replaced by our WaveGC. Our main target is to boost the base models by this replacement.

### D.9 Hyper-parameters Settings

We implement our WaveGC in PyTorch, and list some important parameter values in our model in Table 17. Please note that for the five long-range datasets, we follow the parameter budget $\sim$500k (Dwivedi et al., 2022).

### D.10 Operating Environment

The environment where our code runs is shown as follows:

Table 17: The values of parameters used in WaveGC (T: True; F: False).

| Method | Dataset | # parameters | $\rho$ | $J$ | $\overline{s}$ | Tight frames | Parameters sharing | $\aleph$ |
|---|---|---|---|---|---|---|---|---|
| Transformer +WaveGC | CS | 437k | 3 | 3 | {0.5, 0.5, 0.5} | T | T | 0.1 |
| | Photo | 122k | 3 | 3 | {1.0, 1.0, 1.0} | T | T | 0.1 |
| | Computer | 150k | 7 | 3 | {1.0, 1.0, 1.0} | T | T | 0.1 |
| | CoraFull | 547k | 3 | 3 | {2.0, 2.0, 2.0} | T | T | 0.1 |
| | ogbn-arxiv | 2,091k | 3 | 3 | {0.1, 0.8, 5.0} | F | T | 0.03 |
| | PascalVOC-SP | 477k | 3 | 3 | {0.5, 1.0, 10.0} | T | F | / |
| | PCQM-Contact | 480k | 5 | 3 | {0.5, 1.0, 5.0} | T | F | / |
| | COCO-SP | 553k | 3 | 3 | {10.0, 10.0, 10.0} | T | T | / |
| | Peptides-func | 467k | 5 | 3 | {10.0, 10.0, 10.0} | T | T | / |
| | Peptides-struct | 534k | 2 | 3 | {0.3, 1.0, 10.0} | F | F | / |
| SAN +WaveGC | CS | 524k | 3 | 3 | {0.5, 0.5, 0.5} | T | T | 0.1 |
| | Photo | 262k | 3 | 3 | {1.0, 1.0, 1.0} | T | T | 0.01 |
| | Computer | 292k | 3 | 3 | {2.0, 2.0, 2.0} | T | T | 0.1 |
| | CoraFull | 619k | 3 | 2 | {2.0, 2.0} | T | T | 0.1 |
| | ogbn-arxiv | 2,352k | 3 | 3 | {0.1, 0.8, 5.0} | F | T | 0.03 |
| | PascalVOC-SP | 464k | 3 | 3 | {0.5, 1.0, 10.0} | T | F | / |
| | PCQM-Contact | 411k | 3 | 3 | {0.5, 1.0, 5.0} | T | F | / |
| | COCO-SP | 469k | 3 | 3 | {10.0, 10.0, 10.0} | T | T | / |
| | Peptides-func | 405k | 5 | 3 | {10.0, 10.0, 10.0} | T | T | / |
| | Peptides-struct | 406k | 3 | 3 | {0.3, 1.0, 10.0} | T | F | / |
| GraphGPS +WaveGC | CS | 495k | 3 | 3 | {0.5, 0.5, 0.5} | T | T | 0.1 |
| | Photo | 136k | 3 | 3 | {1.0, 1.0, 1.0} | T | T | 0.1 |
| | Computer | 167k | 3 | 3 | {1.0, 1.0, 1.0} | T | T | 0.1 |
| | CoraFull | 621k | 3 | 3 | {2.0, 2.0, 2.0} | T | T | 0.1 |
| | ogbn-arxiv | 2,354k | 3 | 3 | {0.1, 0.8, 5.0} | F | T | 0.03 |
| | PascalVOC-SP | 506k | 5 | 3 | {0.5, 1.0, 10.0} | T | F | / |
| | PCQM-Contact | 508k | 5 | 3 | {0.5, 1.0, 5.0} | T | F | / |
| | COCO-SP | 546k | 3 | 3 | {0.5, 1.0, 10.0} | T | F | / |
| | Peptides-func | 496k | 5 | 3 | {10.0, 10.0, 10.0} | T | T | / |
| | Peptides-struct | 534k | 3 | 3 | {0.3, 1.0, 10.0} | F | F | / |

- Operating system: Linux version 5.11.0-43-generic

- CPU information: Intel(R) Xeon(R) Gold 6226R CPU @ 2.90GHz

- GPU information: NVIDIA RTX A5000

# E    RELATED WORK

**Graph Transformer.** Graph Transformer (GT) has attracted considerable attentions, where researchers mainly focus on two aspects, i.e., positional encoding and reduction of computational complexity. Firstly, a suitable positional encoding (PE) can assist the Transformer to understand the topology and complex relationships within the graph. GT (Dwivedi & Bresson, 2020) proposes to employ Laplacian eigenvectors as PE with randomly flipping their signs. Graphormer (Ying et al., 2021) takes the distance of the shortest path between two nodes as spatial encoding, which is involved in attention calculation as a bias. SAN (Kreuzer et al., 2021) conducts a learned positional encoding architecture to address key limitations of previous GT analyzed in the paper. GraphGPS (Rampásek et al., 2022) provides different choices for PE, consisting of LapPE, RWSE, SignNet and Equiv-StableLapPE. GRIT (Ma et al., 2023) uses the proposed relative random walk probabilities (RRWP) initial PE to incorporate graph inductive biases. (Geisler et al., 2023) proposes two direction- and structure-aware PE for directed graphs, i.e., Magnetic Laplacian and directional random walk encoding. Both GraphTrans (Wu et al., 2021) and SAT (Chen et al., 2022a) adopts a GNN cascaded with Transformer, where GNN can be viewed as an implicit PE to capture the local structure. Secondly, because of the huge complexity of attention computation $O(N^2)$, some studies endeavor to reduce it to the linear complexity. ANS-GT (Zhang et al., 2022) proposes a hierarchical attention scheme with graph coarsening. DIFFORMER (Wu et al., 2023a) introduces an energy constrained diffusion model with a linear-complexity version. EXPHORMER (Shirzad et al., 2023) consists of a sparse attention mechanism based on virtual global nodes and expander graphs. NAGphormer (Chen et al., 2022b) can be trained in a mini-bath manner by aggregating neighbors from different hops with Hop2Token. NodeFormer (Wu et al., 2022) enables the efficient computation via kernerlized Gumbel-Softmax operator. SGFormer (Wu et al., 2023b) is empowered by a simple attention model that can efficiently propagate information among arbitrary nodes. Besides these two main aspects, Edgeformers (Jin

et al., 2023) and EGT (Hussain et al., 2022) additionally explore the edges by injecting edge text information or designing residual edge channels respectively. Recently, Xing et al. (2024) are the first to reveal the over-globalizing problem in graph transformer, and propose CoBFormer to improve the GT capacity on local modeling with a theoretical guarantee. Furthermore, BRAIN NETWORK TRANSFORMER (Kan et al., 2022) and Grover (Rong et al., 2020) explore the applications of GT on human brains and molecular data.

**Graph Wavelet Transform.** Graph wavelet transform is a generalization of classical wavelet transform c(Mallat, 1999) into graph domain. SGWT (Hammond et al., 2011) defines the computing paradigm on weighted graph via spectral graph theory. Specifically, it defines scaling operation in time field as the scaling on eigenvalues. The authors also prove the localization properties of SGWT in the spatial domain in the limit of fine scales. To accelerate the computation on transform, they additionally present a fast Chebyshev polynomial approximation algorithm. Following SGWT, there have been some efforts on designing more powerful graph wavelet bases in spectral domain, whereas these methods have different flaws. GWNN (Xu et al., 2019a) chooses heat kernel as the filter to construct the bases. SGWF (Shen et al., 2021) proposes an end-to-end learned kernel function using MLP. LGWNN (Xu et al., 2022) designs neural network-parameterized lifting structures, where the lifting operation is based on diffusion wavelets. FGT (Zheng et al., 2022) introduces a decimated framelet for multiscale representation and constructs an up-down coarse-grained chain. SpGAT **??** introduces the attention mechanism in the spectral domain, using diffusion kernel as a basis. The graph wavelet bases learnt from these five methods are not guaranteed as band-pass filters in $\lambda \in [0, 2]$ and thus violate admissibility condition (Mallat, 1999). UFGCONV (Zheng et al., 2021) defines a framelet-based graph convolution with Haar-type filters. Wave-GD (Cho et al., 2023) focuses on graph generation with score-based diffusion, and realizes multiple resolutions with graph wavelet. Furthermore, the authors set $k(s) = sxe^{-sx}$ as band-pass filter and $k(s) = e^{-sx}$ as low-pass filter. WaveNet (Yang et al., 2024) relies on Haar wavelets as bases, and uses the highest-order scaling function to approximate all the other wavelets and scaling functions. WGGP (Opolka et al., 2022) integrates Gaussian processes with Mexican Hat to represent varying levels of smoothness on the graph. The above four methods fix the form of the constructed wavelets, extremely limiting the adaptivity to different datasets. In this paper, our WaveGC constructs band-pass filter and low-pass filter purely depending on the even terms and odd terms of Chebyshev polynomials. In this case, the admissibility condition is strictly guaranteed, and the constructed graph wavelets can be arbitrarily complex and flexible with the number of truncated terms increasing. In addition, there are also some papers exploring the applications of graph wavelets on multi-resolution matrix factorization (Hy & Kondor, 2022) and tensor decomposition (Leonardi & Van De Ville, 2013). SEA-GWNN (Deb et al., 2024) focuses on the second generation of wavelets, or lifting schemes, which is a different topic from ours.

