# OpenReview forum: "Wavelet-based Graph Convolution via Chebyshev Decomposition"
_ICLR.cc/2025/Conference — ICLR 2025 Conference Withdrawn Submission_

### Official Review · Reviewer_P13U · 2024-10-29

**Soundness:** 2
**Presentation:** 3
**Contribution:** 2
**Rating:** 3
**Confidence:** 5

**Summary:**

The paper proposes WaveGC, a wavelet-based graph convolution network that enhances spectral graph convolution by capturing both short- and long-range information with improved flexibility. Using multi-resolution spectral bases and a matrix-valued filter kernel, WaveGC outperforms traditional graph convolutional networks and Transformers. The proposed method for generating graph wavelets ensures strict wavelet admissibility, and experiments show consistent performance improvements across tasks when WaveGC replaces Transformer components.

**Strengths:**

1. Using odd and even Chebyshev bases to approximate wavelet bases is an interesting approach that satisfies theoretical requirements.
2. This paper presents a comprehensive theoretical and empirical analysis, demonstrating the promise and effectiveness of the proposed method.
3. The writing is clear, well-structured, and easy to follow.

**Weaknesses:**

1. The complexity of WaveGC poses a significant limitation. Polynomial-based convolutions, such as GPRGNN and ChebNetII, are readily scalable to large graphs like Papers100M. In contrast, WaveGC not only requires eigendecomposition but also exhibits quadratic complexity, making training on billion-scale graphs impractical.
2. Using odd and even Chebyshev bases to approximate functions g and h appears suboptimal, as relying solely on odd or even bases limits the ability to approximate arbitrary constrained functions effectively.
3. WaveGC introduces substantial complexity. Compared to polynomial-based convolutions, it requires more preprocessing operations and additional parameters yet achieves only comparable performance, as illustrated in Tables 3 and 4. A more detailed comparison of preprocessing time, training time, and memory usage with the latest baselines—not solely with the basic method, such as Transformers—would enhance the analysis.
4. While the authors emphasize that WaveGC captures both short- and long-range information, the improvements reported in Table 4 are marginal. A broader discussion and comparison with the latest methods, such as Subgraphormer[1], PolyFormer[2], and [3], would be valuable.

[1] Bar-Shalom, Guy, Beatrice Bevilacqua, and Haggai Maron. "Subgraphormer: Unifying Subgraph GNNs and Graph Transformers via Graph Products." In ICML, 2024.
[2] Ma, Jiahong, Mingguo He, and Zhewei Wei. "PolyFormer: Scalable Node-wise Filters via Polynomial Graph Transformer." In KDD. 2024.
[3] Cai, Chen, et al. "On the connection between mpnn and graph transformer." In ICML, 2023.

**Questions:**

1. Please begin by clarifying the weaknesses section.
2. Do the authors consider using top-k eigendecomposition or randomized SVD to manage computational costs? How does this impact performance?
3. Is it possible to approximate arbitrary g and h functions in Equation 5 using only odd or even terms? Would this approach be suboptimal?

---

### Official Review · Reviewer_zMyE · 2024-10-30

**Soundness:** 3
**Presentation:** 3
**Contribution:** 3
**Rating:** 5
**Confidence:** 5

**Summary:**

The paper proposes a novel wavelet-based graph convolution network, termed WaveGC, which utilizes multi-resolution spectral bases and a matrix-valued filter kernel. The method integrates wavelet-based filtering with graph convolution, aiming to improve the model’s ability to capture both short- and long-range dependencies in graph data. The authors present theoretical proofs demonstrating WaveGC’s flexibility in handling diverse receptive fields, distinguishing it from traditional graph convolution networks and graph Transformers. Experimental results on multiple benchmarks show that WaveGC consistently outperforms existing models across various datasets, particularly in tasks involving both local and global node interactions.

**Strengths:**

* The paper introduces a novel integration of wavelet bases and graph convolution, specifically utilizing Chebyshev decomposition. This approach enhances the model’s ability to capture both short- and long-range dependencies, providing greater flexibility in graph signal processing compared to traditional methods.

* The authors present rigorous theoretical analysis, demonstrating WaveGC’s capability to handle diverse receptive fields.

* WaveGC consistently outperforms existing models across multiple benchmarks, demonstrating its effectiveness in tasks requiring both local and global node interactions.

**Weaknesses:**

* The authors should further clarify how WaveGC offers substantial advancements beyond prior work, particularly in terms of practical contributions and algorithmic novelty.

* The comparison with baselines could be more comprehensive. Although WaveGC shows improvements over certain models, additional comparisons on heterophilic graphs are expected.

* The experimental setup and parameter selection lack sufficient details, particularly regarding hyperparameter tuning, which may impact reproducibility.

**Questions:**

* The experimental comparisons focus on Transformer-based models. Could the authors include more comparisons with recent graph convolutional models that also emphasize multi-resolution filtering (such as spectral-based methods like Zheng et al., 2021; Cho et al., 2023)?

* What are the key differences between the approach of using Chebyshev decomposition for constructing wavelets in this paper and the method described in Dong, B. (2017) 'Sparse Representation on Graphs by Tight Wavelet Frames and Applications' (Applied and Computational Harmonic Analysis, 42(3), 452-479)? The techniques appear similar; could the authors clarify any distinct technical innovations or advantages introduced in this work?

* How interpretable are the learned wavelet bases and matrix-valued kernels in WaveGC? Are there ways to visualize the learned filters to understand how the model processes short- and long-range information differently? Any further insights?

---

### Official Review · Reviewer_sU3j · 2024-11-03

**Soundness:** 2
**Presentation:** 3
**Contribution:** 2
**Rating:** 5
**Confidence:** 3

**Summary:**

The paper introduces WaveGC, a novel wavelet-based graph convolution network designed to enhance data filtering on graphs. While previous methods have primarily focused on standard Fourier transforms and vector-valued spectral functions, they often lack the flexibility needed to model signal distributions across extensive spatial ranges. WaveGC incorporates multi-resolution spectral bases and a matrix-valued filter kernel, allowing it to effectively capture and separate short-range and long-range information.

**Strengths:**

1. The theoretical support of the proposed WaveGC on capturing the short-range and long-range information.

2. The comprehensive experiments on both short-range and long-range datasets.

**Weaknesses:**

1. From the experimental results, the proposed WaveGC does not achieve better performance compared to other graph Transformers (GTs) on short-range datasets.

2. On long-range datasets, the improvements observed on some datasets, such as VOC and PS, are minor.

3. The computational complexity of the proposed method, especially on large-scale datasets is vey large.

4. There are some types and errors in the paper.  For example, the bold and underline highlight on Ps in Table 4.

**Questions:**

No

---

### Official Review · Reviewer_8Jrp · 2024-11-04

**Soundness:** 3
**Presentation:** 2
**Contribution:** 2
**Rating:** 5
**Confidence:** 4

**Summary:**

In essence, this paper proposed a graph convolution operation based on wavelets (WaveGC), and analyzed its theoretical capability to capture information at short and long ranges. Numerical simulations evaluate its performance.

**Strengths:**

-This paper proposed a new wavelet-based graph convolution network, which integrates multi-resolution spectral bases and a matrix-valued filter kernel. Theoretical analysis seems to be sound and the method appears to show better performance than baselines in considered datasets.
- The paper is well written overall.

**Weaknesses:**

While the paper seems to have some merits, this reviewer is left with the following concerns:
1.	The computational cost and the message exchange overhead of the WaveGC are not clear to the reviewer. I noticed that the authors list the computation as limitation. I would suggest to provide details about computation and message exchange requirements.
2.	In Section 2, the SGWT is given in (2) using only unit wavelet basis \Psi. However, what is the relationship between SGWT and the scaling function basis \Phi? \Phi does not show up in SGWT but is introduced in this section. The interplay between \Psi and \Phi should be stated clearly here, which is confusing to the readers now, especially those less familiar with spectral graph wavelets.
3.	What is the motivation of selecting the formation of (6) for learnable parameters? This should be stated clearly.
4.	Can you provide the interpretation of the mathematical expression of maximal mixing? What does this definition mean intuitively?
5.	For experiments, how do the authors select the hyperparameters of the model? I guess the selection will influence the performance quite significantly.
6.	For comparisons with baselines in Table 3 and Table 4, why are the considered baselines different in Table 3 and Table 4? I think the baselines should be consistent for both short-range and long-range tasks.
7.	Similarly, for comparisons with existing spectral methods in table 5, why do the authors only consider “Computer” and “VOC”? I think the tasks should be consistent to (same as) the other comparisons in Table 3 and Table 4, especially because comparisons with existing spectral methods is important from the reviewer’s perspective.
8.	The last concern is that the performance improvement compared to baselines seems to be incremental (not impressive), I would like to see the computational cost comparisons of WaveGC with baselines, like training time and inference time, for a more comprehensive comparison.
9. Several works on graph scattering transforms have not been cited nor compared against, among which [R1,R2]. As such the contribution of with work w.r.t. earlier literature is not well positioned.
[R1] Gama, F., Ribeiro, A., & Bruna, J. (2018). Diffusion scattering transforms on graphs. arXiv preprint arXiv:1806.08829.
[R2]Cheng, X., Chen, X., & Mallat, S. (2016). Deep Haar scattering networks. Information and Inference: A Journal of the IMA, 5(2), 105-133.

**Questions:**

see weaknesses part

---

### Note · Authors · 2024-11-14

I have read and agree with the venue's withdrawal policy on behalf of myself and my co-authors.